# MATCHER: SEGMENT ANYTHING WITH ONE SHOT USING ALL-PURPOSE FEATURE MATCHING

**Yang Liu**[1*†] **Muzhi Zhu**[1*] **Hengtao Li**[1*] **Hao Chen**[1] **Xinlong Wang**[2] **Chunhua Shen**[1]

[1] Zhejiang University, China
`{yangliu9610,zhumuzhi,liht,haochen.cad,chunhuashen}@zju.edu.cn`

[2] Beijing Academy of Artificial Intelligence
`wangxinlong@baai.ac.cn`

## ABSTRACT

Powered by large-scale pre-training, vision foundation models exhibit significant potential in open-world image understanding. However, unlike large language models that excel at directly tackling various language tasks, vision foundation models require a task-specific model structure followed by fine-tuning on specific tasks. In this work, we present **Matcher**, a novel perception paradigm that utilizes off-the-shelf vision foundation models to address various perception tasks. Matcher can segment anything by using an in-context example without training. Additionally, we design three effective components within the Matcher framework to collaborate with these foundation models and unleash their full potential in diverse perception tasks. Matcher demonstrates impressive generalization performance across various segmentation tasks, all without training. For example, it achieves $52.7\%$ mIoU on COCO-$20^i$ with one example, surpassing the state-of-the-art specialist model by $1.6\%$. In addition, Matcher achieves $33.0\%$ mIoU on the proposed LVIS-$92^i$ for one-shot semantic segmentation, outperforming the state-of-the-art generalist model by $14.4\%$. Our visualization results further showcase the open-world generality and flexibility of Matcher when applied to images in the wild. Our code is at: https://github.com/aim-uofa/Matcher

## 1 INTRODUCTION

Pre-trained on web-scale datasets, large language models (LLMs) (Brown et al., 2020; Ouyang et al., 2022; Chowdhery et al., 2022; Zhang et al., 2022b; Zeng et al., 2022; Touvron et al., 2023), like ChatGPT (OpenAI, 2023), have revolutionized natural language processing (NLP). These foundation models (Bommasani et al., 2021) show remarkable transfer capability on tasks and data distributions beyond their training scope. LLMs demonstrate powerful zero-shot and few-shot generalization (Brown et al., 2020) and solve various language tasks well, *e.g.*, language understanding, generation, interaction, and reasoning.

Research of vision foundation models (VFMs) is catching up with NLP. Driven by large-scale image-text contrastive pre-training, CLIP (Radford et al., 2021) and ALIGN (Jia et al., 2021) perform strong zero-shot transfer ability to various classification tasks. DINOv2 (Oquab et al., 2023) demonstrates impressive visual feature matching ability by learning to capture complex information at the image and pixel level from raw image data alone. Recently, the Segment Anything Model (SAM) (Kirillov et al., 2023) has achieved impressive class-agnostic segmentation performance by training on the SA-1B dataset, including 1B masks and 11M images. Unlike LLMs (Brown et al., 2020; Touvron et al., 2023), which seamlessly incorporate various language tasks through a unified model structure and pre-training method, VFMs face limitations when directly addressing diverse perception tasks. For example, these methods often require a task-specific model structure followed by fine-tuning on a specific task (He et al., 2022; Oquab et al., 2023).

---

*Equal contribution. †Part of the work was done when YL was an intern at Beijing Academy of Artificial Intelligence. CS is the corresponding author.

In this work, we aim to find a new visual research paradigm: *investigating the utilization of VFMs for effectively addressing a wide range of perception tasks*, *e.g.*, semantic segmentation, part segmentation, and video object segmentation, *without training*. Using foundation models is non-trivial due to the following challenges: 1) Although VFMs contain rich knowledge, it remains challenging to directly leverage individual models for downstream perception tasks. Take SAM as an example. While SAM can perform impressive zero-shot class-agnostic segmentation performance across various tasks, it cannot provide the semantic categories for the predicted masks. Besides, SAM prefers to predict multiple ambiguous mask outputs. It is difficult to select the appropriate mask as the final result for different tasks. 2) Various tasks involve complex and diverse perception requirements. For example, semantic segmentation predicts pixels with the same semantics. However, video object segmentation needs to distinguish individual instances within those semantic categories. Additionally, the structural distinctions of different tasks need to be considered, encompassing diverse semantic granularities ranging from individual parts to complete entities and multiple instances. Thus, naively combining the foundation models can lead to subpar performance.

To address these challenges, we present **Matcher**, a novel perception framework that effectively incorporates different foundation models for tackling diverse perception tasks by using a single in-context example. We draw inspiration from the remarkable generalization capabilities exhibited by LLMs in various NLP tasks through in-context learning (Brown et al., 2020). Prompted by the in-context example, Matcher can understand the specific task and utilizes DINOv2 to locate the target by matching the corresponding semantic feature. Subsequently, leveraging this coarse location information, Matcher employs SAM to predict accurate perceptual results. In addition, we design three effective components within the Matcher framework to collaborate with foundation models and fully unleash their potential in diverse perception tasks. First, we devise a bidirectional matching strategy for accurate cross-image semantic dense matching and a robust prompt sampler for mask proposal generation. This strategy increases the diversity of mask proposals and suppresses fragmented false-positive masks induced by matching outliers. Furthermore, we perform instance-level matching between the reference mask and mask proposals to select high-quality masks. We utilize three effective metrics, *i.e.*, *emd*, *purity*, and *coverage*, to estimate the mask proposals based on semantic similarity and the quality of the mask proposals, respectively. Finally, by controlling the number of merged masks, Matcher can produce controllable mask output to instances of the same semantics in the target image.

Our comprehensive experiments demonstrate that Matcher has superior generalization performance across various segmentation tasks, all without the need for training. For one-shot semantic segmentation, Matcher achieves **52.7%** mIoU on COCO-$20^i$ (Nguyen & Todorovic, 2019), surpassing the state-of-the-art specialist model by **1.6%**, and achieves **33.0%** mIoU on the proposed LVIS-$92^i$, outperforming the state-of-the-art generalist model SegGPT (Wang et al., 2023b) by **14.4%**. And Matcher outperforms concurrent PerSAM (Zhang et al., 2023) by a large margin (+**29.2%** mean mIoU on COCO-$20^i$, +**11.4%** mIoU on FSS-1000 (Li et al., 2020), and +**10.7%** mean mIoU on LVIS-$92^i$), suggesting that depending solely on SAM limits the generalization capabilities for semantically-driven tasks, *e.g.*, semantic segmentation. Moreover, evaluated on two proposed benchmarks, Matcher shows outstanding generalization on one-shot object part segmentation tasks. Specifically, Matcher outperforms other methods by about **10.0%** mean mIoU on both benchmarks. Matcher also achieves competitive performance for video object segmentation on both DAVIS 2017 val (Pont-Tuset et al., 2017) and DAVIS 2016 val (Perazzi et al., 2016). In addition, exhaustive ablation studies verify the effectiveness of the proposed components of Matcher. Finally, our visualization results show robust generality and flexibility never seen before.

Our main contributions are summarized as follows:

- We present Matcher, one of the first perception frameworks for exploring the potential of vision foundation models in tackling diverse perception tasks, *e.g.*, one-shot semantic segmentation, one-shot object part segmentation, and video object segmentation.

- We design three components, *i.e.*, bidirectional matching, robust prompt sampler, and instance-level matching, which can effectively unleash the ability of vision foundation models to improve both the segmentation quality and open-set generality.

- Our comprehensive results demonstrate the impressive performance and powerful generalization of Matcher. Sufficient ablation studies show the effectiveness of the proposed components.

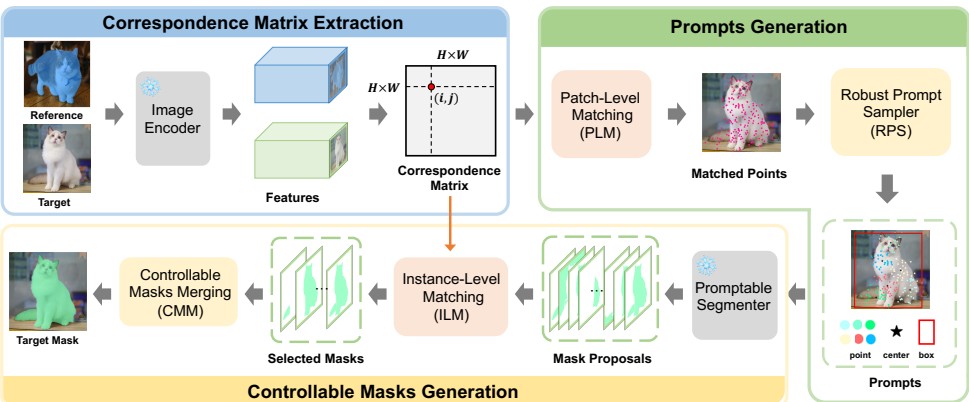

Figure 1: An overview of Matcher. Our training-free framework addresses various segmentation tasks through three operations: Correspondence Matrix Extraction, Prompts Generation, and Controllable Masks Generation.

## 2 RELATED WORK

**Vision Foundation Models** Powered by large-scale pre-training, vision foundation models have achieved great success in computer vision. Motivated by masked language modeling (Devlin et al., 2019; Liu et al., 2019) in natural language processing, MAE (He et al., 2022) uses an asymmetric encoder-decoder and conducts masked image modeling to effectively and efficiently train scalable vision Transformer (Dosovitskiy et al., 2020) models. CLIP (Radford et al., 2021) learns image representations from scratch on 400 million image-text pairs and demonstrates impressive zero-shot image classification ability. By performing image and patch level discriminative self-supervised learning, DINOv2 (Oquab et al., 2023) learns all-purpose visual features for various downstream tasks. Recently, pre-trained with 1B masks and 11M images, Segment Anything Model (SAM) (Kirillov et al., 2023) emerges with impressive zero-shot class-agnostic segmentation performance. Although vision foundation models have shown exceptional fine-tuning performance, they have limited capabilities in various visual perception tasks. However, large language models (Brown et al., 2020; Chowdhery et al., 2022; Touvron et al., 2023), like ChatGPT (OpenAI, 2023), can solve a wide range of language tasks without training. Motivated by this, this work shows that various perception tasks can be solved training-free by utilizing off-the-shelf vision foundation models to perform in-context inference.

**Vision Generalist for Segmentation** Recently, a growing effort has been made to unify various segmentation tasks under a single model using Transformer architecture (Vaswani et al., 2017). The generalist Painter (Wang et al., 2023a) redefines the output of different vision tasks as images and utilizes masked image modeling on continuous pixels to perform in-context training with supervised datasets. As a variant of Painter, SegGPT (Wang et al., 2023b) introduces a novel random coloring approach for in-context training to improve the model's generalization ability. By prompting spatial queries, *e.g.*, points, and text queries, *e.g.*, textual prompts, SEEM (Zou et al., 2023) performs various segmentation tasks effectively. More recently, PerSAM and PerSAM-F (Zhang et al., 2023) adapt SAM for personalized segmentation and video object segmentation without training or with two trainable parameters. This work presents Matcher, a training-free framework for segmenting anything with one shot. Unlike these methods, Matcher demonstrates impressive generalization performance across various segmentation tasks by integrating different foundation models.

## 3 METHOD

Matcher is a training-free framework that segments anything with one shot by integrating an all-purpose feature extraction model (*e.g.*, DINOv2 (Oquab et al., 2023))and a class-agnostic segmentation model (*e.g.*, SAM (Kirillov et al., 2023)). For the given in-context example, including reference image $\mathbf{x}_r$ and mask $m_r$, Matcher can segment the objects or parts of a target image $\mathbf{x}_t$ with the same semantics. The overview of Matcher is depicted in Fig. 1. Our framework consists of three components: Correspondence Matrix Extraction (CME), Prompts Generation (PG), and Controllable

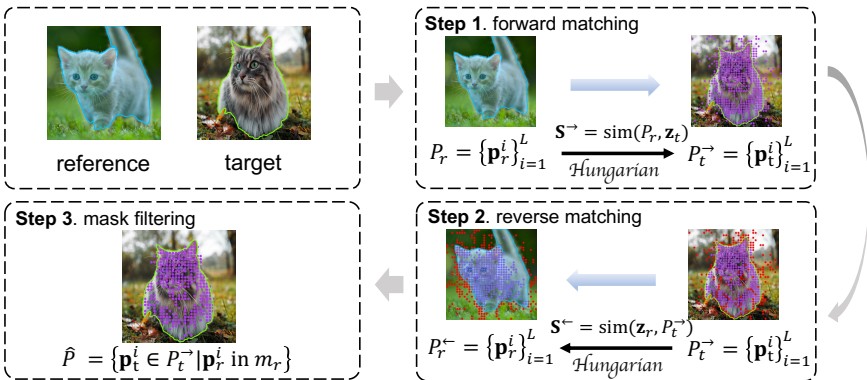

Figure 2: Illustration of the proposed bidirectional matching. Bidirectional matching consists of three steps: forward matching, reverse matching, and mask filtering. Purple points denote the matched points. Red points denote the outliers.

Masks Generation (CMG). First, Matcher extracts a correspondence matrix by calculating the similarity between the image features of $\mathbf{x}_r$ and $\mathbf{x}_t$. Then, we conduct patch-level matching, followed by sampling multiple groups of prompts from the matched points. These prompts serve as inputs to SAM, enabling the generation of mask proposals. Finally, we perform an instance-level matching between the reference mask and mask proposals to select high-quality masks. We elaborate on the three components in the following subsections.

## 3.1 CORRESPONDENCE MATRIX EXTRACTION

We rely on off-the-self image encoders to extract features for both the reference and target images. Given inputs $\mathbf{x}_r$ and $\mathbf{x}_t$, the encoder outputs patch-level features $\mathbf{z}_r, \mathbf{z}_t \in \mathbb{R}^{H \times W \times C}$. Patch-wise similarity between the two features is computed to discovery the best matching regions of the reference mask on the target image. We define a correspondence matrix $\mathbf{S} \in \mathbb{R}^{HW \times HW}$ as follows,

$$(\mathbf{S})_{ij} = \frac{\mathbf{z}_r^i \cdot \mathbf{z}_t^j}{\|\mathbf{z}_r^i\| \cdot \|\mathbf{z}_t^j\|}, \tag{1}$$

where $(\mathbf{S})_{ij}$ denotes the cosine similarity between $i$-th patch feature $\mathbf{z}_r^i$ of $\mathbf{z}_r$ and $j$-th patch feature $\mathbf{z}_t^j$ of $\mathbf{z}_t$. We can denote the above formulation in a compact form as $\mathbf{S} = \mathrm{sim}(\mathbf{z}_r, \mathbf{z}_t)$.

Ideally, the matched patches should have the highest similarity. This could be challenging in practice, since the reference and target objects could have different appearances or even belong to different categories. This requires the encoder to embed rich and detailed information in these features.

## 3.2 PROMPTS GENERATION

Given the dense correspondence matrix, we can get a coarse segmentation mask by selecting the most similar patches in the target image. However, this naive approach leads to inaccurate, fragmented result with many outliers. Hence, we use the correspondence feature to generate high quality point and box guidance for promptable segmentation. The process involves a bidirectional patch matching and a diverse prompt sampler.

**Patch-Level Matching**  The encoder tends to produce wrong matches in hard cases such as ambiguous context and multiple instances. We propose a bidirectional matching strategy to eliminate the matching outliers.

- As shown in Fig. 2, we first perform bipartite matching between the points on the reference mask $P_r = \{\mathbf{p}_r^i\}_{i=1}^L$ and $\mathbf{z}_t$ to obtain the forward matched points on the target image $P_t^{\rightarrow} = \{\mathbf{p}_t^i\}_{i=1}^L$ using the forward correspondence matrix $\mathbf{S}^{\rightarrow} = \mathrm{sim}(P_r, \mathbf{z}_t)$.

- Then, we perform another bipartite matching, named the reverse matching between $P_t^{\rightarrow}$ and $\mathbf{z}_r$ to obtain the reverse matched points on the reference image $P_r^{\leftarrow} = \{\mathbf{p}_r^i\}_{i=1}^L$ using the reverse correspondence matrix $\mathbf{S}^{\leftarrow} = \mathrm{sim}(\mathbf{z}_r, P_t^{\rightarrow})$.

- Finally, we filter out the points in the forward set if the corresponding reverse points are not on the reference mask $m_r$. The final matched points are $\hat{P} = \{\mathbf{p}_t^i \in P_t^{\rightarrow} | \mathbf{p}_r^i \text{ in } m_r\}$.

**Robust Prompt Sampler** Inspired by the effective prompt-engineering (Kojima et al., 2022; Wei et al., 2022; Li & Liang, 2021; Zhu et al., 2023), we introduce a robust prompt sampler for the promptable segmenter to support robust segmentation with various semantic granularity, from parts and whole to multiple instances. We first cluster the matched points $\hat{P}$ based on their locations into $K$ clusters $\hat{P}_k$ with $k$-means++ (Arthur & Vassilvitskii, 2007). Then the following three types of subsets are sampled as prompts:

- Part-level prompts are sampled within each cluster $P^p \subset \hat{P}_k$;
- Instance-level prompts are sampled within all matched points $P^i \subset \hat{P}$;
- Global prompts are sampled within the set of cluster centers $P^g \subset C$ to encourage coverage, where $C = \{c_1, c_2, \ldots, c_k\}$ are the cluster centers.

In practice, we find this strategy not only increases the diversity of mask proposals but also suppresses fragmented false-positive masks induced by matching outliers.

## 3.3 CONTROLLABLE MASKS GENERATION

The edge features of an object extracted by the image encoder can confuse background information, inducing some indistinguishable outliers. These outliers can generate some false-positive masks. To overcome this difficulty, we further select high-quality masks from the mask proposals via an instance-level matching module and then merge the selected masks to obtain the final target mask.

**Instance-Level Matching** We perform the instance-level matching between the reference mask and mask proposals to select great masks. We formulate the matching to the Optimal Transport (OT) problem and employ the Earth Mover's Distance (EMD) to compute a structural distance between dense semantic features inside the masks to determine mask relevance. The cost matrix of the OT problem can be calculated by $\mathbf{C} = \frac{1}{2}(1 - \mathbf{S})$. We use the method proposed in (Bonneel et al., 2011) to calculate the EMD, noted as *emd*.

In addition, we propose two other mask proposal metrics, *i.e.*, $purity = \frac{Num(\hat{P}_{mp})}{Area(m_p)}$ and $coverage = \frac{Num(\hat{P}_{mp})}{Num(\hat{P})}$, to assess the quality of the mask proposals simultaneously, where $\hat{P}_{mp} = \{\mathbf{p}_t^i \in P_t^{\rightarrow} | \mathbf{p}_t^i \text{ in } m_p\}$, $Num(\cdot)$ represents the number of points, $Area(\cdot)$ represents the area of the mask, and $m_p$ is the mask proposal. A higher degree of *purity* promotes the selection of part-level masks, while a higher degree of *coverage* promotes the selection of instance-level masks. The false-positive mask fragments can be filtered using the proposed metrics through appropriate thresholds, followed by a score-based selection process to identify the top-k highest-quality masks

$$score = \alpha \cdot (1 - emd) + \beta \cdot purity \cdot coverage^\lambda, \tag{2}$$

where $\alpha$, $\beta$, and $\lambda$ are regulation coefficients between different metrics. By manipulating the number of merged masks, Matcher can produce controllable mask output to instances of the same semantics in the target image. More details of *emd*, *purity* and *coverage* are provided in Appendix A.

## 4 EXPERIMENTS

### 4.1 EXPERIMENTS SETTING

**Vision Foundation Models** We use DINOv2 (Oquab et al., 2023) with a ViT-L/14 (Dosovitskiy et al., 2020) as the default image encoder of Matcher. Benefiting from large-scale discriminative self-supervised learning at both the image and patch level, DINOv2 has impressive patch-level representation ability, which promotes exact patch matching between different images. We use the Segment Anything Model (SAM) (Kirillov et al., 2023) with ViT-H as the segmenter of Matcher. Pre-trained with 1B masks and 11M images, SAM emerges with impressive zero-shot segmentation performance. Combining these vision foundation models has the enormous potential to touch open-world image understanding. **In all experiments, we do not perform any training for the Matcher.** More implementation details are provided in Appendix B.

| Methods | Venue | COCO-20$^i$ | | FSS-1000 | | LVIS-92$^i$ | |
|---|---|---|---|---|---|---|---|
| | | one-shot | few-shot | one-shot | few-shot | one-shot | few-shot |
| *specialist model* | | | | | | | |
| HSNet (Min et al., 2021) | ICCV'21 | 41.2 | 49.5 | 86.5 | 88.5 | 17.4 | 22.9 |
| VAT (Hong et al., 2022) | ECCV'22 | 41.3 | 47.9 | 90.3 | 90.8 | 18.5 | 22.7 |
| FPTrans (Zhang et al., 2022a) | NeurIPS'22 | 47.0 | 58.9 | - | - | - | - |
| MSANet (Iqbal et al., 2022) | arXiv'22 | 51.1 | 56.8 | - | - | - | - |
| *generalist model* | | | | | | | |
| Painter (Wang et al., 2023a) | CVPR'23 | 33.1 | 32.6 | 61.7 | 62.3 | 10.5 | 10.9 |
| SegGPT (Wang et al., 2023b) | ICCV'23 | 56.1 | 67.9 | 85.6 | 89.3 | 18.6 | 25.4 |
| PerSAM[†‡] (Zhang et al., 2023) | arXiv'23 | 23.0 | - | 71.2 | - | 11.5 | - |
| PerSAM-F[‡] | | 23.5 | - | 75.6 | - | 12.3 | - |
| Matcher[†‡] | this work | **52.7** | **60.7** | **87.0** | **89.6** | **33.0** | **40.0** |

Table 1: Results of few-shot semantic segmentation on COCO-20$^i$, FSS-1000, and LVIS-92$^i$. Gray indicates the model is trained by in-domain datasets. † indicates the training-free method. ‡ indicates the method using SAM. Note that the training data of SegGPT includes COCO.

## 4.2 FEW-SHOT SEMANTIC SEGMENTATION

**Datasets** For few-shot semantic segmentation, we evaluate the performance of Matcher on COCO-20$^i$ (Nguyen & Todorovic, 2019), FSS-1000 (Li et al., 2020), and LVIS-92$^i$. COCO-20$^i$ partitions the 80 categories of the MSCOCO dataset (Lin et al., 2014) into four cross-validation folds, each containing 60 training classes and 20 test classes. FSS-1000 consists of mask-annotated images from 1,000 classes, with 520, 240, and 240 classes in the training, validation, and test sets, respectively. We verify Matcher on the test sets of COCO-20$^i$ and FSS-1000 following the evaluation scheme of (Min et al., 2021). Note that, different from specialist models, we do not train Matcher on these datasets. In addition, based on the LVIS dataset (Gupta et al., 2019), we create LVIS-92$^i$, a more challenging benchmark for evaluating the generalization of a model across datasets. After removing the classes with less than two images, we retained a total of 920 classes for further analysis. These classes were then divided into 10 equal folds for testing purposes. For each fold, we randomly sample a reference image and a target image for evaluation and conduct 2,300 episodes.

**Results** We compare the Matcher against a variety of specialist models, such as HSNet (Min et al., 2021), VAT (Hong et al., 2022), FPTrans (Zhang et al., 2022a), and MSANet (Iqbal et al., 2022), as well as generalist models like Painter (Wang et al., 2023a), SegGPT (Wang et al., 2023b), and PerSAM (Zhang et al., 2023). As shown in Table 1, for COCO-20$^i$, Matcher achieves **52.7%** and **60.7%** mean mIoU with one-shot and few-shot, surpassing the state-of-the-art specialist models MSANet and achieving comparable with SegGPT. Note that the training data of SegGPT includes COCO. For FSS-1000, Matcher exhibits highly competitive performance compared with specialist models and surpasses all generalist models. Furthermore, Matcher outperforms training-free PerSAM and fine-tuning PerSAM-F by a significant margin (**+29.2%** mean mIoU on COCO-20$^i$, **+11.4%** mIoU on FSS-1000, and **+10.7%** mean mIoU on LVIS-92$^i$), suggesting that depending solely on SAM results in limited generalization capabilities for semantic tasks. For LVIS-92$^i$, we compare the cross-dataset generalization abilities of Matcher and other models. For specialist models, we report the average performance of four pre-trained models on COCO-20$^i$. Matcher achieves **33.0%** and **40.0%** mean mIoU with one-shot and few-shot, outperforming the state-of-the-art generalist model SegGPT by **14.4%** and **14.6%**. Our results indicate that Matcher exhibits robust generalization capabilities that are not present in the other models.

## 4.3 ONE-SHOT OBJECT PART SEGMENTATION

**Datasets** Requiring a fine-grained understanding of objects, object part segmentation is a more challenging task than segmenting an object. We build two benchmarks to evaluate the performance of Matcher on one-shot part segmentation, *i.e.*, PASCAL-Part and PACO-Part. Based on PASCAL VOC 2010 (Everingham et al., 2010) and its body part annotations (Chen et al., 2014), we build the PASCAL-Part dataset following (Morabia et al., 2020). The dataset consists of four superclasses, *i.e.*, animals, indoor, person, and vehicles. There are five subclasses for animals, three for indoor, one for person, and six for vehicles. There are 56 different object parts in total. PACO (Ramanathan et al., 2023) is a newly released dataset that provides 75 object categories and 456 object part categories. Based on the PACO dataset, we build the more difficult PACO-Part benchmark for one-shot object part segmentation. We filter the object parts whose area is minimal and those with less than two images, resulting in 303 remaining object parts. We split these parts into four folds, each with about

| Methods | Venue | PASCAL-Part | | | | | PACO-Part | | | | |
|---|---|---|---|---|---|---|---|---|---|---|---|
| | | animals | indoor | person | vehicles | mean | F0 | F1 | F2 | F3 | mean |
| HSNet (Min et al., 2021) | ICCV'21 | 21.2 | 53.0 | 20.2 | 35.1 | 32.4 | 20.8 | 21.3 | 25.5 | 22.6 | 22.6 |
| VAT (Hong et al., 2022) | ECCV'22 | 21.5 | 55.9 | 20.7 | 36.1 | 33.6 | 22.0 | 22.9 | 26.0 | 23.1 | 23.5 |
| Painter (Wang et al., 2023a) | CVPR'23 | 20.2 | 49.5 | 17.6 | 34.4 | 30.4 | 13.7 | 12.5 | 15.0 | 15.1 | 14.1 |
| SegGPT (Wang et al., 2023b) | ICCV'23 | 22.8 | 50.9 | 31.3 | 38.0 | 35.8 | 13.9 | 12.6 | 14.8 | 12.7 | 13.5 |
| PerSAM[†‡] (Zhang et al., 2023) | arXiv'23 | 19.9 | 51.8 | 18.6 | 32.0 | 30.1 | 19.4 | 20.5 | 23.8 | 21.2 | 21.2 |
| Matcher[†‡] | this work | **37.1** | **56.3** | **32.4** | **45.7** | **42.9** | **32.7** | **35.6** | **36.5** | **34.1** | **34.7** |

Table 2: Results of one-shot part segmentation on PASCAL-Part and PACO-Part. † indicates the training-free method. ‡ indicates the method using SAM.

| Methods | Venue | DAVIS 2017 val | | | DAVIS 2016 val | | |
|---|---|---|---|---|---|---|---|
| | | $J\&F$ | $J$ | $F$ | $J\&F$ | $J$ | $F$ |
| *with video data* | | | | | | | |
| AGAME (Johnander et al., 2019) | CVPR'19 | 70.0 | 67.2 | 72.7 | - | - | - |
| AGSS (Lin et al., 2019) | ICCV'19 | 67.4 | 64.9 | 69.9 | - | - | - |
| AFB-URR (Liang et al., 2020) | NeurIPS'20 | 74.6 | 73.0 | 76.1 | - | - | - |
| AOT (Yang et al., 2021) | NeurIPS'21 | 85.4 | 82.4 | 88.4 | 92.0 | 90.7 | 93.3 |
| SWEM (Lin et al., 2022) | CVPR'22 | 84.3 | 81.2 | 87.4 | 91.3 | 89.9 | 92.6 |
| XMem (Cheng & Schwing, 2022) | ECCV'22 | 87.7 | 84.0 | 91.4 | 92.0 | 90.7 | 93.2 |
| *without video data* | | | | | | | |
| Painter (Wang et al., 2023a) | CVPR'23 | 34.6 | 28.5 | 40.8 | 70.3 | 69.6 | 70.9 |
| SegGPT (Wang et al., 2023b) | ICCV'23 | 75.6 | 72.5 | 78.6 | 83.7 | 83.6 | 83.8 |
| PerSAM[†‡] (Zhang et al., 2023) | arXiv'23 | 60.3 | 56.6 | 63.9 | - | - | - |
| PerSAM-F[‡] | | 71.9 | 69.0 | 74.8 | - | - | - |
| Matcher[†‡] | this work | **79.5** | **76.5** | **82.6** | **86.1** | **85.2** | **86.7** |

Table 3: Results of video object segmentation on DAVIS 2017 val, and DAVIS 2016 val. Gray indicates the model is trained on target datasets with video data. † indicates the training-free method. ‡ indicates the method using SAM.

76 different object parts. We crop all objects out with their bounding box to evaluate the one-shot part segmentation on both two datasets. More details are provided in Appendix C.

**Results** We compare our Matcher with HSNet, VAT, Painter, and PerSAM. For HSNet and VAT, we use the models pre-trained on PASCAL-$5^i$ (Shaban et al., 2017) and COCO-$20^i$ for PASCAL-Part and PACO-Part, respectively. As shown in Table 2, the results demonstrate that Matcher outperforms all previous methods by a large margin. Specifically, Matcher outperforms the SAM-based PerSAM +**12.8%** mean mIoU on PASCAL-Part and +**13.5%** on PACO-Part, respectively. SAM has shown the potential to segment any object into three levels: whole, part, and subpart (Kirillov et al., 2023). However, it cannot distinguish these ambiguity masks due to the lack of semantics. This suggests that SAM alone cannot work well on one-shot object part segmentation. Our method empowers SAM for semantic tasks by combining it with an all-purpose feature extractor and achieves effective generalization performance on fine-grained object part segmentation tasks with an in-context example.

### 4.4 VIDEO OBJECT SEGMENTATION

**Datasets** Video object segmentation (VOS) aims to segment a specific object in video frames. Following Wang et al. (2023b), we evaluate Matcher on the validation split of two datasets, *i.e.*, DAVIS 2017 val (Pont-Tuset et al., 2017), and DAVIS 2016 val (Perazzi et al., 2016), under the semi-supervised VOS setting. Two commonly used metrics in VOS, the $J$ score and the $F$ score, are used for evaluation.

**Details** In order to track particular moving objects in a video, we maintain a reference memory containing features and the intermediate predictions of the previous frames in Matcher. We determine which frame to retain in the memory according to the *score* (see subsection 3.3) of the frames. Considering that objects are more likely to be similar to those in adjacent frames, we apply a decay ratio decreasing by time to the *score*. We fix the given reference image and mask in the memory to avoid failing when some objects disappear in intermediate frames and reappear later.

**Results** We compare Matcher with the models trained with or without video data on different datasets in Table 3. The results show that Matcher can achieve competitive performance compared with the models trained with video data. Moreover, Matcher outperforms the models trained without video data, *e.g.*, SegGPT and PerSAM-F, on both two datasets. These results suggest that Matcher can effectively generalize to VOS tasks without training.

| ILM | COCO-20$^i$ mean mIoU | FSS-1000 mIoU | DAVIS 2017 $J\&F$ |
|---|---|---|---|
| | 29.0 | 76.2 | 39.9 |
| ✓ | **52.7** | **87.0** | **79.5** |

(a) Ablation study of ILM.

| Strategy | COCO-20$^i$ mean mIoU | FSS-1000 mIoU | DAVIS 2017 $J\&F$ |
|---|---|---|---|
| forward | 50.6 | 81.1 | 73.5 |
| reverse | 21.4 | 47.7 | 41.3 |
| bidirectional | **52.7** | **87.0** | **79.5** |

(b) Ablation study of bidirectional matching.

| emd | p&c | COCO-20$^i$ mean mIoU | FSS-1000 mIoU | DAVIS 2017 $J\&F$ |
|---|---|---|---|---|
| ✓ | | 51.3 | 86.3 | 67.5 |
| | ✓ | 35.3 | 86.3 | 76.3 |
| ✓ | ✓ | **52.7** | **87.0** | **79.5** |

(c) Effect of different mask proposal metrics.

| Frames | DAVIS 2017 | | | |
|---|---|---|---|---|
| | 1 | 2 | 4 | 6 |
| $J\&F$ | 73.5 | 74.4 | **79.5** | 78.0 |
| $J$ | 70.0 | 70.5 | **76.5** | 74.9 |
| $F$ | 77.5 | 78.2 | **82.6** | 81.1 |

(d) Effect of the number of frames for VOS.

Table 4: Ablation study. We report the mean mIoU of four folds on COCO-20$^i$, mIoU on FSS-1000, and $J\&F$ on DAVIS 2017 val. Default setting settings are marked in Gray .

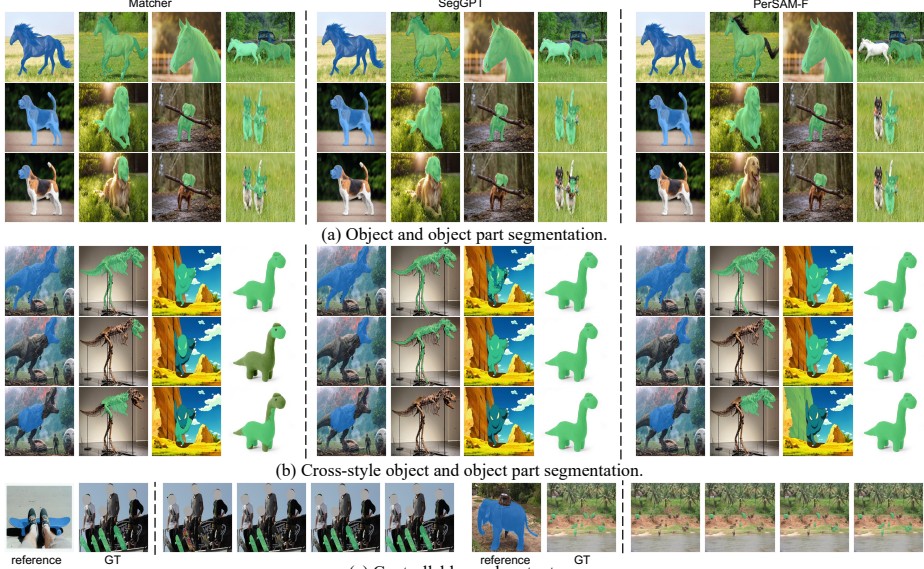

(a) Object and object part segmentation.

(b) Cross-style object and object part segmentation.

(c) Controllable mask output.

Figure 3: Qualitative results of one-shot segmentation.

## 4.5 ABLATION STUDY

As shown in Table 4, we conduct ablation studies on both the difficult COCO-20$^i$ dataset and the simple FSS-1000 dataset for one-shot semantic segmentation and DAVIS 2017 val for video object segmentation to sufficiently verify the effectiveness of our proposed components. In this subsection, we explore the effects of matching modules (ILM), patch-level matching strategies, and different mask proposal metrics.

**Ablation Study of ILM** Patch-level matching (PLM) and instance-level matching (ILM) are the vital components of Matcher that bridge the gap between the image encoder and SAM to solve various few-shot perception tasks training-free. As shown in Table 4a, PLM builds the connection between matching and segmenting and empowers Matcher with the capability of performing various few-shot perception tasks training-free. And ILM enhances this capability by a large margin.

**Ablation Study of Bidirectional Matching** As shown in Table 4b, we explore the effects of the forward matching and the reverse matching of the proposed bidirectional matching. For the reverse matching, because the matched points $P_t^{\rightarrow}$ (see subsection 3.2) are unavailable when performing reverse matching directly, we perform the reverse matching between $\mathbf{z}_t$ and $\mathbf{z}_r$. Without the guidance of the reference mask, reverse matching (line 2) produces many wrong matching results, resulting in poor performance. Compared with the forward matching (line 1), our bidirectional matching strategy improves the performance by +2.1% mean mIoU on COCO-20$^i$, by +5.9% mIoU on FSS-1000, and by +6.0% $J\&F$ on DAVIS 2017. These significant improvements show the effectiveness of the proposed bidirectional matching strategy.

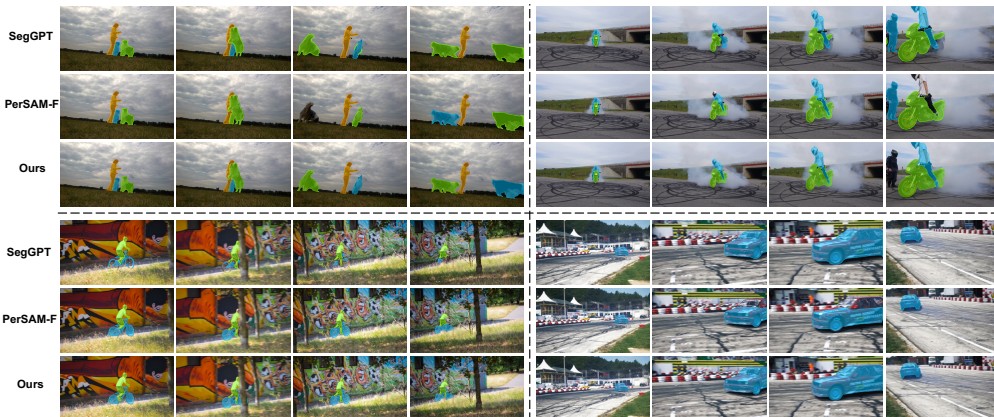

Figure 4: Qualitative results of video object segmentation on DAVIS 2017.

**Ablation Study of Different Mask Proposal Metrics** As shown in Table 4c, *emd* is more effective on the complex COCO-20$^i$ dataset. *emd* evaluates the patch-level feature similarity between the mask proposals and the reference mask that encourages matching all mask proposals with the same category. In contrast, by using *purity* and *coverage*, Matcher can achieve great performance on DAVIS 2017. Compared with *emd*, *purity* and *coverage* are introduced to encourage selecting high-quality mask proposals. Combining these metrics to estimate mask proposals, Matcher can achieve better performance in various segmentation tasks without training.

**Effect of the Number of Frames for VOS** As shown in Table 4d, we also explore the effect of the number of frames on DAVIS 2017 val. The performance of Matcher can be improved as the number of frames increases, and the optimal performance is achieved when using four frames. More ablation studies are provided in Appendix D.

## 4.6 QUALITATIVE RESULTS

To demonstrate the generalization of our Matcher, we visualize the qualitative results of one-shot segmentation in Fig. 3 from three views, *i.e.*, object and object part segmentation, cross-style object and object part segmentation, and controllable mask output. Our Matcher can achieve higher-quality objects and parts masks than SegGPT and PerSAM-F. Better results on cross-style segmentation show the impressive generalization of Matcher due to effective all-feature matching. In addition, by manipulating the number of merged masks, Macther supports multiple instances with the same semantics. Fig. 4 shows qualitative results of VOS on DAVIS 2017. The remarkable results demonstrate that Matcher can effectively unleash the ability of foundation models to improve both the segmentation quality and open-set generality.

## 5 CONCLUSION

In this paper, we present Matcher, a training-free framework integrating off-the-shelf vision foundation models for solving various few-shot segmentation tasks. Combining these foundation models properly leads to positive synergies, and Matcher emerges complex capabilities beyond individual models. The introduced universal components, *i.e.*, bidirectional matching, robust prompt sampler, and instance-level matching, can effectively unleash the ability of these foundation models. Our experiments demonstrate the powerful performance of Matcher for various few-shot segmentation tasks, and our visualization results show open-world generality and flexibility on images in the wild.

**Limitation and Ethics Statement** While Matcher demonstrates impressive performance for semantic-level segmentation, *e.g.*, one-shot semantic segmentation and one-shot object part segmentation, it has relatively limited instance-level matching inherited from the image encoder, which restrains its performance for instance segmentation. However, the comparable VOS performance and the visualization of controllable mask output demonstrate that Matcher has the potential for instance-level segmentation. We will explore it in future work. Our work can unleash the potential of different foundation models for various visual tasks. In addition, our Matcher is built upon open-source foundation models without training, significantly reducing carbon emissions. We do not foresee any obvious undesirable ethical or social impacts now.

ACKNOWLEDGMENTS

This work was supported by National Science and Technology Major Project (No. 2022ZD0118700). The research was in part supported by the Supercomputing Center of Hangzhou City University, which provided advanced computing resources.

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

# APPENDIX

## A  MORE DETAILS OF INSTANCE-LEVEL MATCHING

**The *emd* metric.** The OT problem can be described as follows: suppose that $m$ suppliers $U = \{u_i | i = 1, 2, ..., m\}$ require transport goods for $n$ demanders $D = \{d_j | j = 1, 2, ..., n\}$, where $u_i$ represents the supply units of $i$-th supplier and $d_j$ denotes the demand of $j$-th demanded. The cost of transporting each unit of goods from the $i$-th supplier to the $j$-th demander is represented by $c_{ij}$, and the number of units transported is denoted by $\pi_{ij}$. The goal of the OT problem is to identify a transportation plan $\pi = \{\pi_{ij} | i = 1, ...m, j = 1, ...n\}$ that minimizes the overall transportation cost

$$\min_{\pi} \quad \sum_{i=1}^{m} \sum_{j=1}^{n} c_{ij} \pi_{ij}.$$
$$\text{s.t.} \quad \sum_{j=1}^{n} \pi_{ij} = u_i, \quad \sum_{i=1}^{m} \pi_{ij} = d_j,$$
$$\sum_{i=1}^{m} u_i = \sum_{j=1}^{n} d_j,$$
$$\pi_{ij} \geq 0, \quad i = 1, 2, ...m, \quad j = 1, 2, ...n.$$
(3)

In the context of Matcher, the suppliers are $m$ reference image patches covered by the reference mask, and the demanders are $n$ target image patches covered by the mask proposal (produced by SAM). The goods that the suppliers need to transmit have the same value, *i.e.*, $u_i = \frac{1}{m}, \sum u_i = 1$. Similarly, the goods that the demanders need also have the same value, *i.e.*, $d_j = \frac{1}{n}, \sum d_j = 1$. The cost $c_{ij}$ can be obtained from the cost matrix $\mathbf{C}$ by utilizing the mask proposal $m_p$ and the reference mask $m_r$. Then, we use the method proposed in Bonneel et al. (2011) to calculate the EMD.

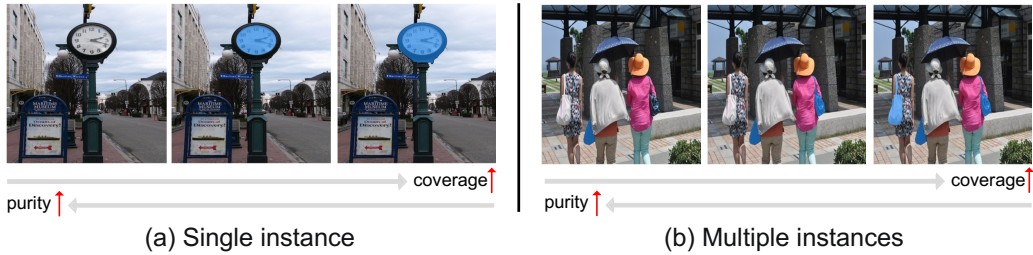

(a) Single instance  (b) Multiple instances

Figure 5: Illustration of the effects of the *purity* and *coverage*.

**The *purity* and *coverage* metrics** Fig. 5 shows examples to demonstrate the effects of the *purity* and *coverage* criteria in two scenarios, *i.e.*, single instance and multiple instances. A higher degree of *purity* promotes the selection of part or single instance masks, while a higher degree of *coverage* promotes the selection of whole or multiple instance masks.

## B  IMPLEMENTATION DETAILS

We use DINOv2 (Oquab et al., 2023) with a ViT-L/14 (Dosovitskiy et al., 2020) as the default image encoder of Matcher. And we use the Segment Anything Model (SAM) (Kirillov et al., 2023) with ViT-H as the segmenter of Matcher. In all experiments, we do not perform any training for the Matcher. We set input image sizes are $518 \times 518$ for one-shot semantic segmentation and object part segmentation and $896 \times 504$ for video object segmentation. We conduct experiments from three semantic granularity for semantic segmentation, *i.e.*, parts (PASCAL-Part and PACO-Part), whole (FSS-1000), and multiple instances (COCO-$20^i$ and LVIS-$92^i$). We set the number of clusters to 8. For COCO-$20^i$ and LVIS-$92^i$, we sample the instance-level points from the matched points and dense image points to encourage SAM to output more instance masks. We set the filtering thresholds *emd* and *purity* to 0.67, 0.02 and set $\alpha$, $\beta$ and $\lambda$ to 1.0, 0.0, and 0.0, respectively. For FSS-1000, we sample the global prompts from centers. We set $\alpha$, $\beta$, and $\lambda$ to 0.8, 0.2, and 1.0, respectively. We sample the points from the matched points and use the smallest axis-aligned box containing these

matched points for PASCAL-Part and PACO-Part. We set the filtering threshold *coverage* to 0.3 and set $\alpha$, $\beta$ and $\lambda$ to 0.5, 0.5, and 0.0, respectively. For video object segmentation, we sample the global prompts from centers. We set the filtering threshold *emd* to 0.75 and set $\alpha$, $\beta$, and $\lambda$ to 0.4, 1.0, and 1.0.

## C  DATASET DETAILS

**PASCAL-Part**  Based on PASCAL VOC 2010 (Everingham et al., 2010) and its body part annotations (Chen et al., 2014), we build the PASCAL-Part dataset following (Morabia et al., 2020). Table 5 shows the part taxonomy of PASCAL-Part dataset. The dataset consists of four superclasses, *i.e.*, animals, indoor, person, and vehicles. There are five subclasses for animals (bird, cat, cow, dog, horse, sheep), three for indoor (bottle, potted plant, tv monitor), one for person (person), and six for vehicles (aeroplane, bicycle, bus, car, motorbike, train). There are 56 different object parts in total.

**PACO-Part**  Based on the PACO (Ramanathan et al., 2023) dataset, we build the more difficult PACO-Part benchmark for one-shot object part segmentation. Firstly, we filter the categories having only 1 sample. Then, we filter low-quality examples with an extremely small pixel area within PACO, which leads to significant noise during evaluation, resulting in 303 remaining object parts. Table 6 shows the part taxonomy of the PACO-Part dataset. We split these parts into four folds, each with about 76 different object parts.

| Superclasses | Subclasses | Parts |
|---|---|---|
| animals | bird | face, leg, neck, tail, torso, wings |
| | cat | face, leg, neck, tail, torso |
| | cow | face, leg, neck, tail, torso |
| | dog | face, leg, neck, tail, torso |
| | horse | face, leg, neck, tail, torso |
| | sheep | face, leg, neck, tail, torso |
| indoor | bottle | body |
| | potted plant | plant, pot |
| | tv monitor | screen |
| person | person | face, arm & hand, leg, neck, torso |
| vehicles | aeroplane | body, engine, wheel, wings |
| | bicycle | wheel |
| | bus | door, vehicle side, wheel, windows |
| | car | door, vehicle side, wheel, windows |
| | motorbike | wheel |
| | train | train coach, train head |

Table 5: Part taxonomy of PASCAL-Part

## D  ADDITIONAL RESULTS AND ANALYSIS

**Effect of Different Image Encoders**  Table 7a shows the comparison experiments of CLIP, MAE, and DINOv2. DINOv2 achieves the best performance on all datasets. Because the text-image contrastive pre-training limits learning complex pixel-level information, CLIP cannot precisely match image patches. Although MAE can extract pixel-level features by masked image modeling, it performs poorly. We suspect that the patch-level features extracted by MAE confuse the information about the surrounding patches, resulting in mistaken feature matching. In contrast, pre-trained by image-level and patch-level discriminative self-supervised learning, DIVOv2 extracts all-purpose visual features and exhibit impressive patch-level feature matching ability. As a training-free general perception framework, Matcher can deploy different image encoders. With the continuous development of vision foundation models, the capabilities of vision foundation models will continue to improve, and Matcher's performance and generalization ability will also be enhanced. This is confirmed by the continuous improvement in performance from MAE to CLIP to DINOv2, demonstrating that Matcher has strong flexibility and scalability. Besides, we aim to make Matcher a valuable tool for assessing the performance of pre-trained foundation models on various downstream tasks.

| Fold | Parts |
|------|-------|
| 0 | bench:arm, laptop_computer:back, bowl:base, handbag:base, basket:base, chair:base, glass_(drink_container):base, cellular_telephone:bezel, guitar:body, bucket:body, can:body, soap:body, vase:body, crate:bottom, box:bottom, glass_(drink_container):bottom, basket:bottom, lamp:bulb, television_set:button, watch:case, bottle:closure, book:cover, table:drawer, pillow:embroidery, car_(automobile):fender, dog:foot, bicycle:fork, bicycle:gear, clock:hand, bucket:handle, basket:handle, spoon:handle, bicycle:handlebar, guitar:headstock, sweater:hem, trash_can:hole, bucket:inner_body, hat:inner_side, microwave_oven:inner_side, tray:inner_side, pliers:jaw, laptop_computer:keyboard, shoe:lace, bench:leg, can:lid, fan:light, car_(automobile):mirror, spoon:neck, sweater:neckband, tray:outer_side, bicycle:pedal, can:pull_tab, shoe:quarter, can:rim, mug:rim, pan_(for_cooking):rim, tray:rim, basket:rim, car_(automobile):runningboard, laptop_computer:screen, chair:seat, bicycle:seat_stay, lamp:shade_inner_side, sweater:shoulder, television_set:side, sweater:sleeve, blender:spout, jar:sticker, helmet:strap, table:stretcher, blender:switch, bench:table_top, plastic_bag:text, shoe:tongue, television_set:top, bicycle:top_tube, hat:visor, car_(automobile):wheel, car_(automobile):wiper |
| 1 | chair:apron, chair:back, bench:back, fan:base, cup:base, pan_(for_cooking):base, laptop_computer:base_panel, knife:blade, scissors:blade, bowl:body, sweater:body, handbag:body, mouse_(computer_equipment):body, towel:body, dog:body, bowl:bottom, plate:bottom, television_set:bottom, spoon:bowl, car_(automobile):bumper, cellular_telephone:button, laptop_computer:cable, fan:canopy, bottle:cap, clock:case, pipe:colied_tube, sweater:cuff, microwave_oven:dial, mug:drawing, vase:foot, car_(automobile):grille, plastic_bag:handle, scissors:handle, handbag:handle, mug:handle, cup:handle, pan_(for_cooking):handle, dog:head, bicycle:head_tube, towel:hem, car_(automobile):hood, plastic_bag:inner_body, wallet:inner_body, glass_(drink_container):inner_body, crate:inner_side, pan_(for_cooking):inner_side, plate:inner_wall, soap:label, chair:leg, crate:lid, laptop_computer:logo, broom:lower_bristles, fan:motor, vase:neck, dog:nose, shoe:outsole, lamp:pipe, chair:rail, bucket:rim, bowl:rim, car_(automobile):rim, tape_(sticky_cloth_or_paper):roll, bicycle:saddle, scissors:screw, bench:seat, bicycle:seat_tube, soap:shoulder, box:side, carton:side, earphone:slider, bicycle:stem, chair:stile, bench:stretcher, dog:tail, mug:text, bottle:top, table:top, laptop_computer:touchpad, shoe:vamp, helmet:visor, car_(automobile):window, mouse_(computer_equipment):wire |
| 2 | table:apron, telephone:back_cover, plate:base, kettle:base, blender:base, bicycle:basket, fan:blade, plastic_bag:body, trash_can:body, plate:body, mug:body, kettle:body, towel:border, mug:bottom, telephone:button, microwave_oven:control_panel, microwave_oven:door_handle, dog:ear, helmet:face_shield, scissors:finger_hole, wallet:flap, mirror:frame, kettle:handle, blender:handle, earphone:headband, earphone:housing, bowl:inner_body, trash_can:inner_body, helmet:inner_side, basket:inner_side, calculator:key, bottle:label, mouse_(computer_equipment):left_button, dog:leg, box:lid, trash_can:lid, vase:mouth, pipe:nozzle, slipper_(footwear):outsole, fan:pedestal_column, ladder:rail, hat:rim, plate:rim, trash_can:rim, bottle:ring, car_(automobile):roof, telephone:screen, mouse_(computer_equipment):scroll_wheel, stool:seat, lamp:shade, bottle:shoulder, microwave_oven:side, basket:side, chair:spindle, hat:strap, belt:strap, car_(automobile):taillight, towel:terry_bar, newspaper:text, microwave_oven:time_display, shoe:toe_box, microwave_oven:top, car_(automobile):trunk, slipper_(footwear):vamp, car_(automobile):windowpane, sweater:yoke |
| 3 | chair:arm, remote_control:back, cellular_telephone:back_cover, bottle:base, bucket:base, television_set:base, jar:base, tray:base, lamp:base, telephone:bezel, bottle:body, pencil:body, scarf:body, calculator:body, jar:body, glass_(drink_container):body, bottle:bottom, pan_(for_cooking):bottom, tray:bottom, remote_control:button, bucket:cover, basket:cover, bicycle:down_tube, earphone:ear_pads, dog:eye, guitar:fingerboard, blender:food_cup, stool:footrest, scarf:fringes, knife:handle, vase:handle, car_(automobile):headlight, mug:inner_body, jar:inner_body, cup:inner_body, box:inner_side, carton:inner_side, trash_can:label, table:leg, stool:leg, jar:lid, kettle:lid, car_(automobile):logo, bucket:loop, bottle:neck, dog:neck, pipe:nozzle_stem, book:page, mouse_(computer_equipment):right_button, handbag:rim, jar:rim, glass_(drink_container):rim, cup:rim, cellular_telephone:screen, blender:seal_ring, lamp:shade_cap, table:shelf, crate:side, pan_(for_cooking):side, mouse_(computer_equipment):side_button, chair:skirt, car_(automobile):splashboard, bottle:spout, ladder:step, watch:strap, chair:stretcher, chair:swivel, can:text, jar:text, spoon:tip, slipper_(footwear):toe_box, blender:vapour_cover, chair:wheel, bicycle:wheel, car_(automobile):windshield, handbag:zip |

Table 6: Part taxonomy of PACO-Part

**Effect of different types of prompts** We validated the impact of different prompts on datasets with scenes involving parts (PACO-Part), the whole (FSS-1000), and multiple instances (COCO-20$^i$) in Table 7b: 1) Part-level prompts are needed for PACO-Part, which requires segmenting parts of an instance. However, our experiment results demonstrate that using instance-level prompts yields better results because instance-level prompts cover more situations than part-level prompts. 2) FSS-1000

| Encoder | COCO-20$^i$ mean mIoU | FSS-1000 mIoU | DAVIS 2017 $J\&F$ |
|---|---|---|---|
| MAE | 18.8 | 71.9 | 69.5 |
| CLIP | 32.2 | 77.4 | 73.9 |
| DINOv2 | 52.7 | 87.0 | 79.5 |

(a) Effect of different image encoders.

| Prompts | COCO-20$^i$ | PACO-Part | FSS-1000 |
|---|---|---|---|
| Global | 51.7 | 31.6 | 87.0 |
| Part | 51.7 | 30.2 | 79.2 |
| Instance | 52.7 | 34.0 | 86.5 |

(b) Effect of different types of prompts.

| Segmenter | COCO-20$^i$ | LVIS-92$^i$ | FSS-1000 | PACO-Part |
|---|---|---|---|---|
| SAM | 52.7 | 31.4 | 87.0 | 32.7 |
| Semantic-SAM | 51.1 | 30.1 | 87.5 | 36.0 |

(c) Effect of different segmenters.

| | COCO-20$^i$ | LVIS-92$^i$ | FSS-1000 | PACO-Part |
|---|---|---|---|---|
| Upper Bound | 83.6 | 75.4 | 93.1 | 67.5 |
| Matcher | 52.7 | 31.4 | 87.0 | 32.7 |

(d) Upper bound analysis.

Table 7: Ablation study on the effects of different image encoders, different types of prompts, different segmenters, and upper bound of Matcher.

often involves one instance that occupies the entire image. Thus, global prompts are used for full image coverage. 3) For COCO-20$^i$, which requires detecting all instances in an image, instance-level points are the most effective. All the experiments are conducted on one fold in both three datasets.

| Methods | SAM | DINOv2 | #Params (M) | LVIS-92$^i$ | FSS-1000 |
|---|---|---|---|---|---|
| SegGPT | - | - | 307 | 17.5 | 85.6 |
| Matcher | base | base | 180 | 28.6 | 85.3 |
| | large | base | 399 | 29.9 | 85.7 |
| | large | large | 617 | 30.4 | 86.3 |
| | huge | large | 945 | 31.4 | 87.0 |

(a) Ablation study on model size.

| Number | FSS-1000 |
|---|---|
| 4 | 78.9 |
| 6 | 83.3 |
| 8 | 87.0 |
| 10 | 87.2 |
| 12 | 86.9 |

(b) Ablation study on the cluster number.

Table 8: Ablation study on different model sizes and cluster number.

**Ablation of model size** Table 8a shows the results of Matcher when using VFMs with different model sizes. When using SAM base and DINOv2 base, Matcher still performs well on various datasets and achieves better generalization performance on LVIS-92$^i$ than SegGPT. Besides, as the model size increases, Matcher can continuously improve performance.

**Effect of different segmenters** Table 7c shows the results when using Semantic-SAM (Li et al., 2023) as the segmenter. Semantic-SAM achieves comparable performance with SAM on four benchmarks. Because Semantic-SAM can output more fine-grained masks, it performs better than SAM on PACO-Part. The results indicate that Matcher is a general segmentation framework.

**Upper bound analysis** We conduct experiments on four different datasets and find that the upper bound of Matcher consistently outperforms the current performance on all datasets by a large margin in Table 7d. This indicates that the Matcher framework has more potential. Therefore, Matcher can serve as an effective evaluation criterion for VFMs, assessing the performance of different vision models from a general segmentation perspective. Based on the advantage, Matcher can contribute to developing VFMs.

**How does few-shot segmentation work?** In the few-shot setting, we concatenate multiple references' features and match them with the target image in the PLM. The remaining process is the same as the one-shot setting. Multiple samples provide richer visual details, enabling more accurate matching results and reducing outliers, resulting in performance improvement.

**Visualizations** Fig. 6 shows the quality of background concept segmentation of Matcher. Fig. 7 visualizes the results of Patch-Level Matching, Robust Prompt Sampler and Instance-Level Matching. In addition, We provide more visualizations for one-shot semantic segmentation in Fig. 8, one-shot object part segmentation in Fig. 9 and Fig. 10, controllable mask output in Fig. 11, and video object segmentation in Fig. 12. The remarkable results demonstrate that Matcher can effectively unleash the ability of foundation models to improve both the segmentation quality and open-set generality.

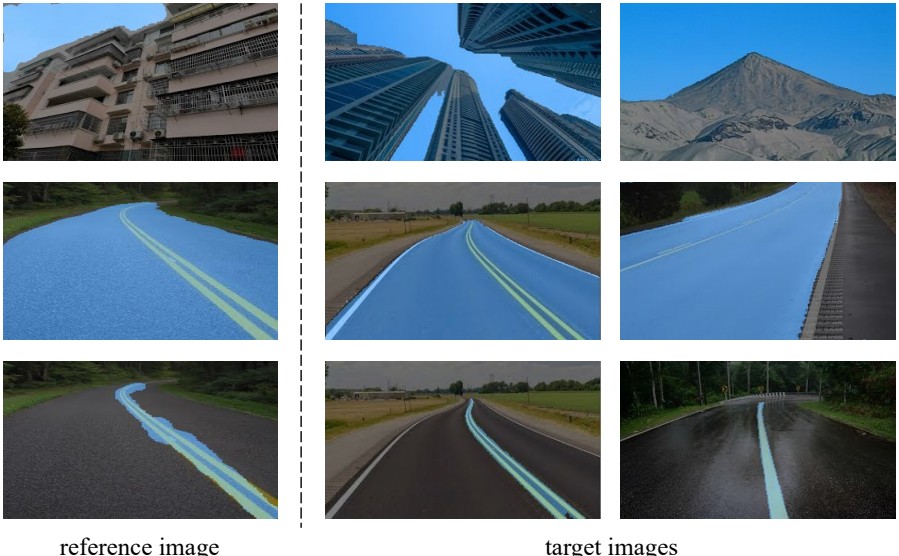

reference image                                    target images

Figure 6: Visualization of Matcher for the quality of background concept segmentation. Matcher can segment various background concepts like SegGPT.

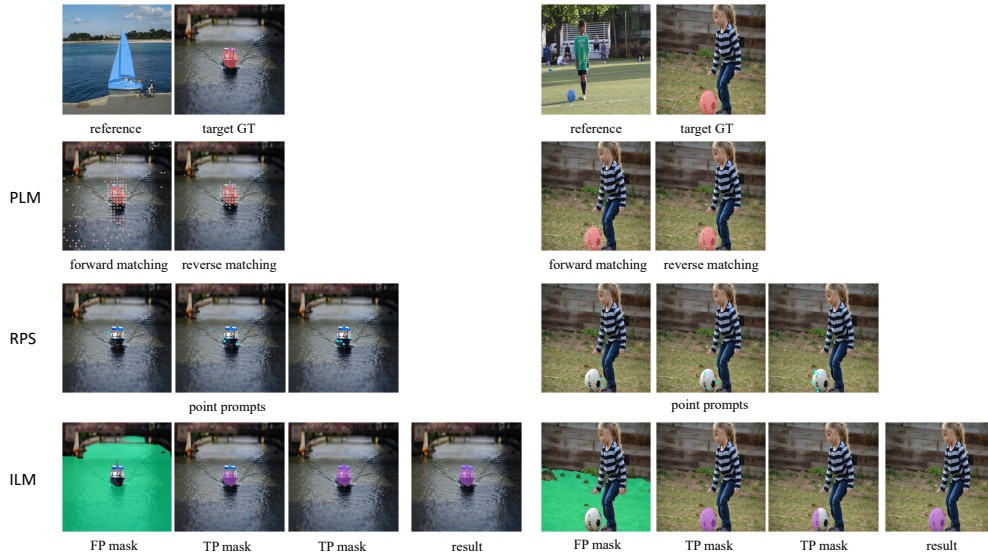

Figure 7: Visualization of the results of Patch-Level Matching (PLM), Robust Prompt Sampler (RPS) and Instance-Level Matching (ILM). (a) For PLM, the Green stars present the correct matched points, and the Red stars present the matched outliers. The PLM can effectively remove most of the outliers via proposed bidirectional matching. (b) RPS can sample various point prompts by using the matched points of PLM. (c) Take the prompts as inputs, SAM can output the mask proposals. Because there are still outliers in the matched points, SAM can output some false-positive (FP) masks. Thus, we propose ILM to filter these FP masks and merge the true-positive (TP) masks. Then, we can get the result. These components within the Matcher framework collaborate with foundation models and unleash their full potential in diverse segmentation tasks.

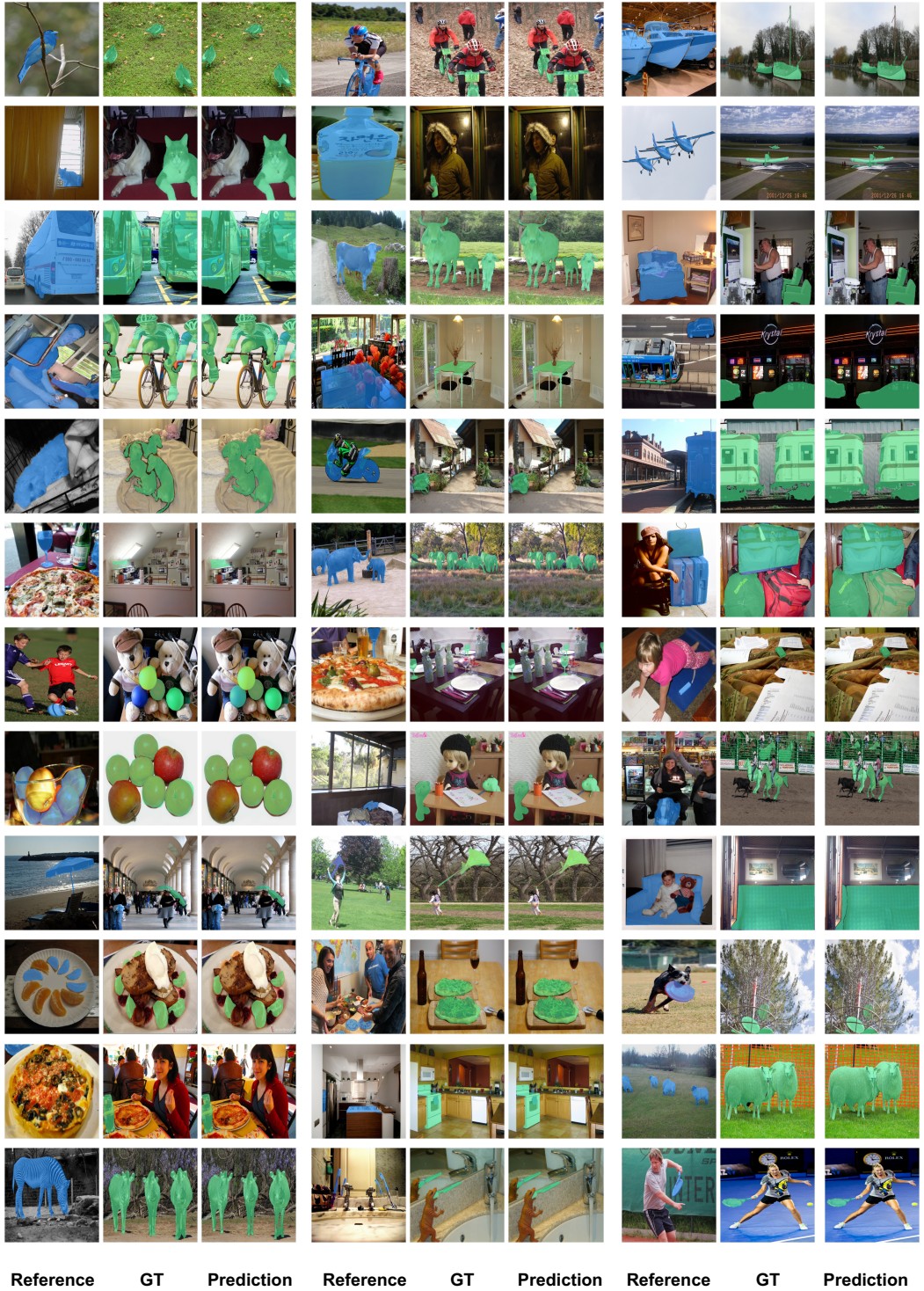

Figure 8: Visualization of one-shot semantic segmentation.

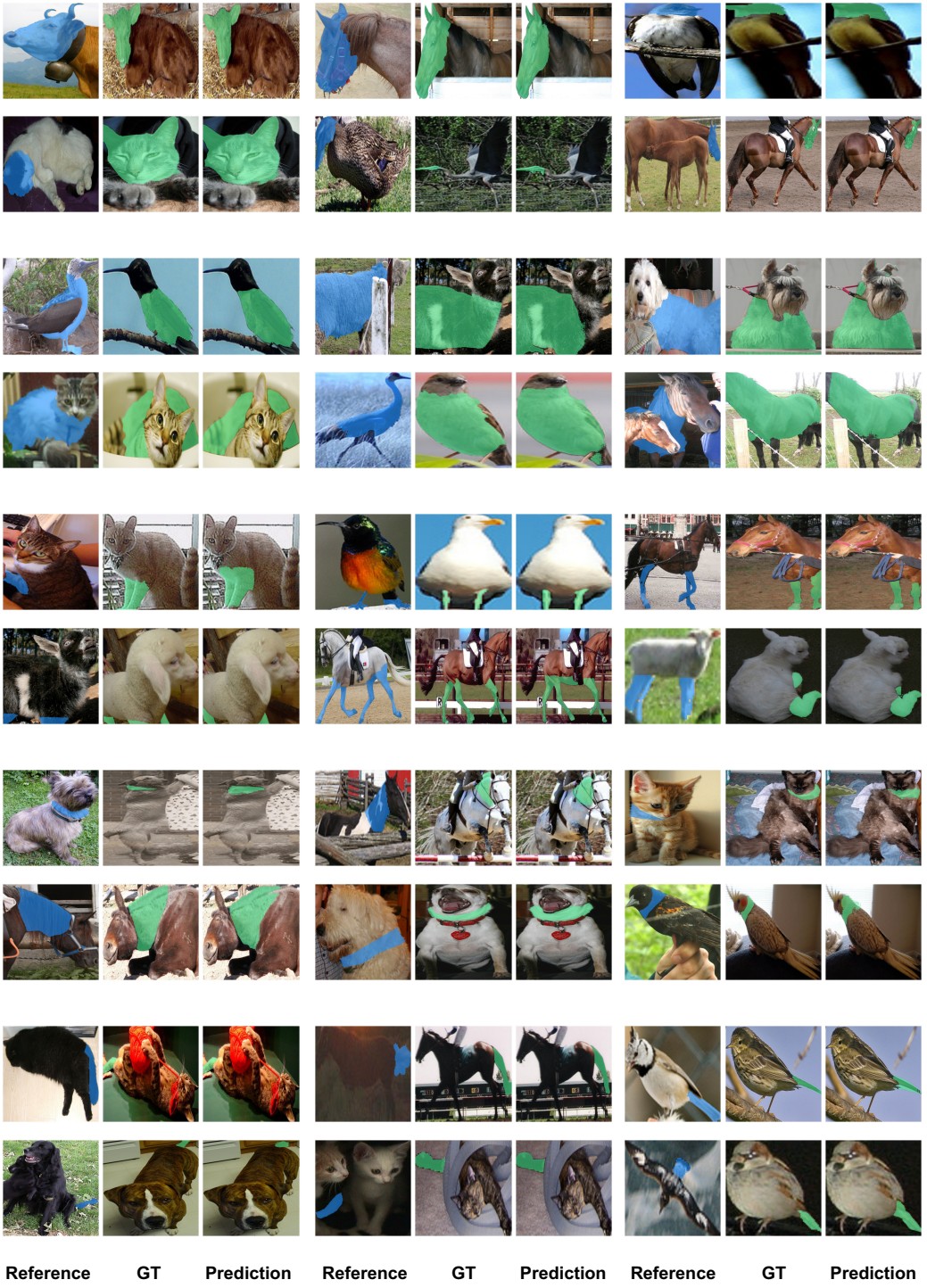

Figure 9: Visualization of one-shot object part segmentation on PASCAL-Part.

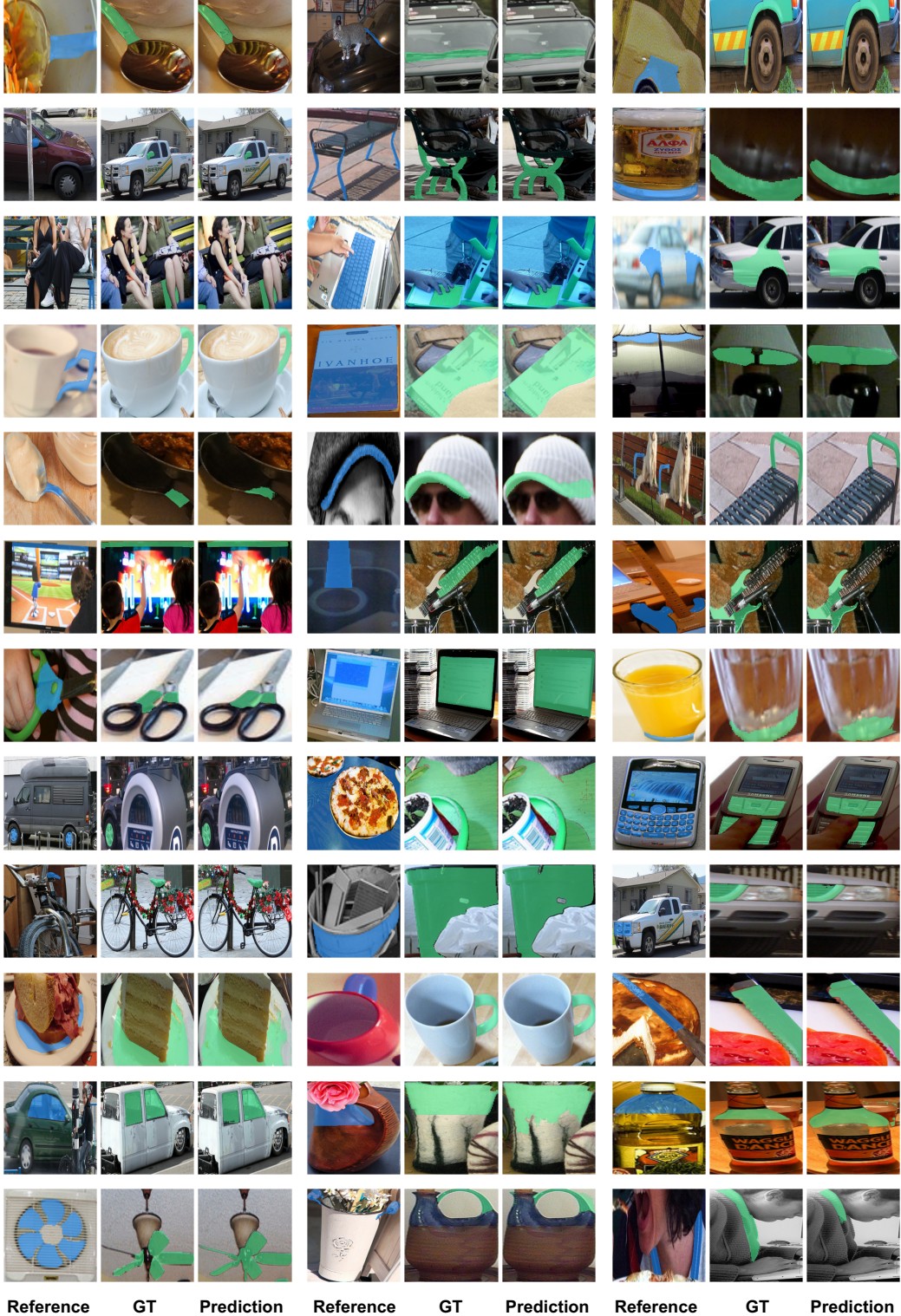

Figure 10: Visualization of one-shot object part segmentation on PACO-Part.

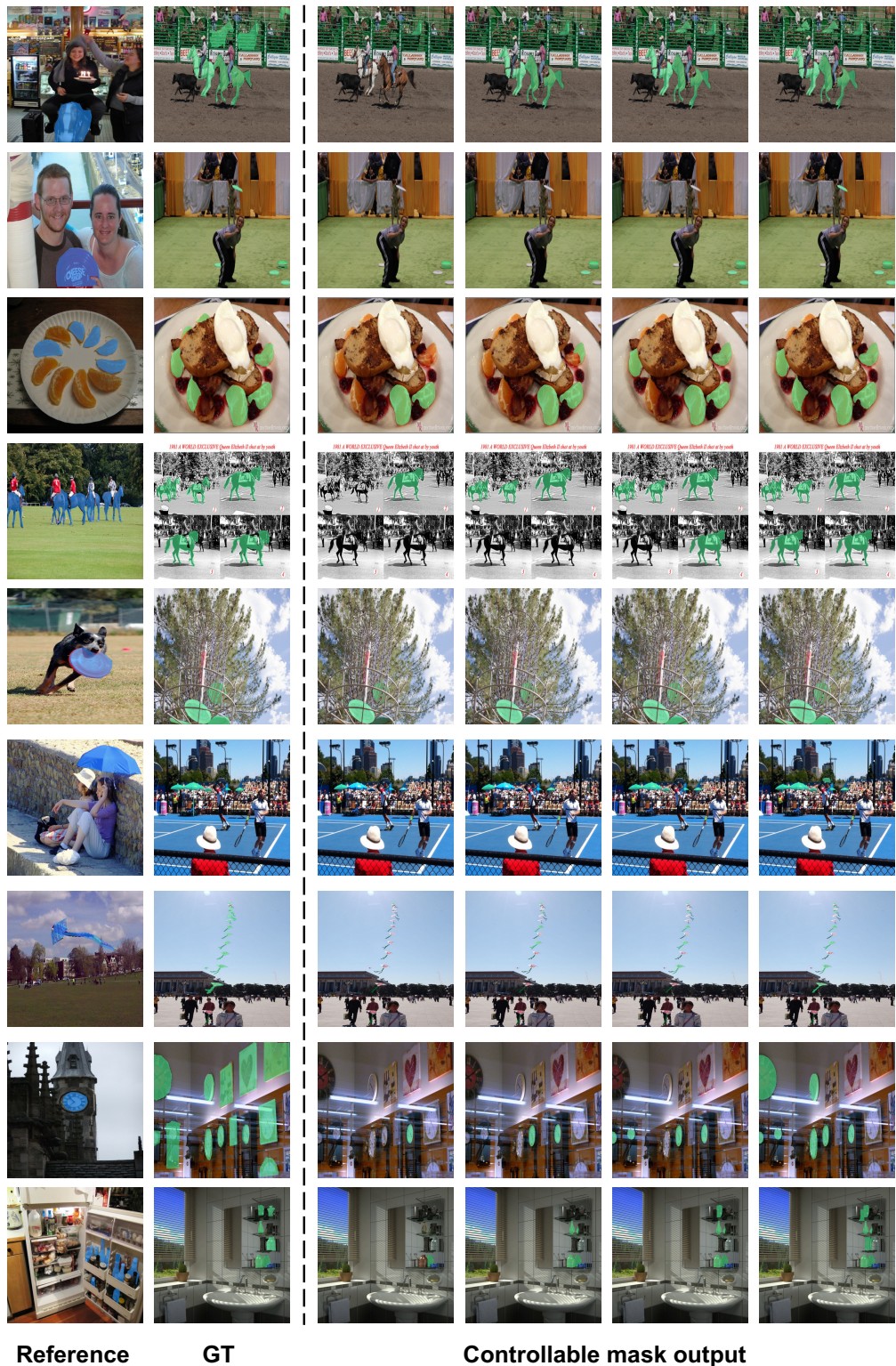

**Reference**          **GT**                    **Controllable mask output**

Figure 11: Visualization of controllable mask output.

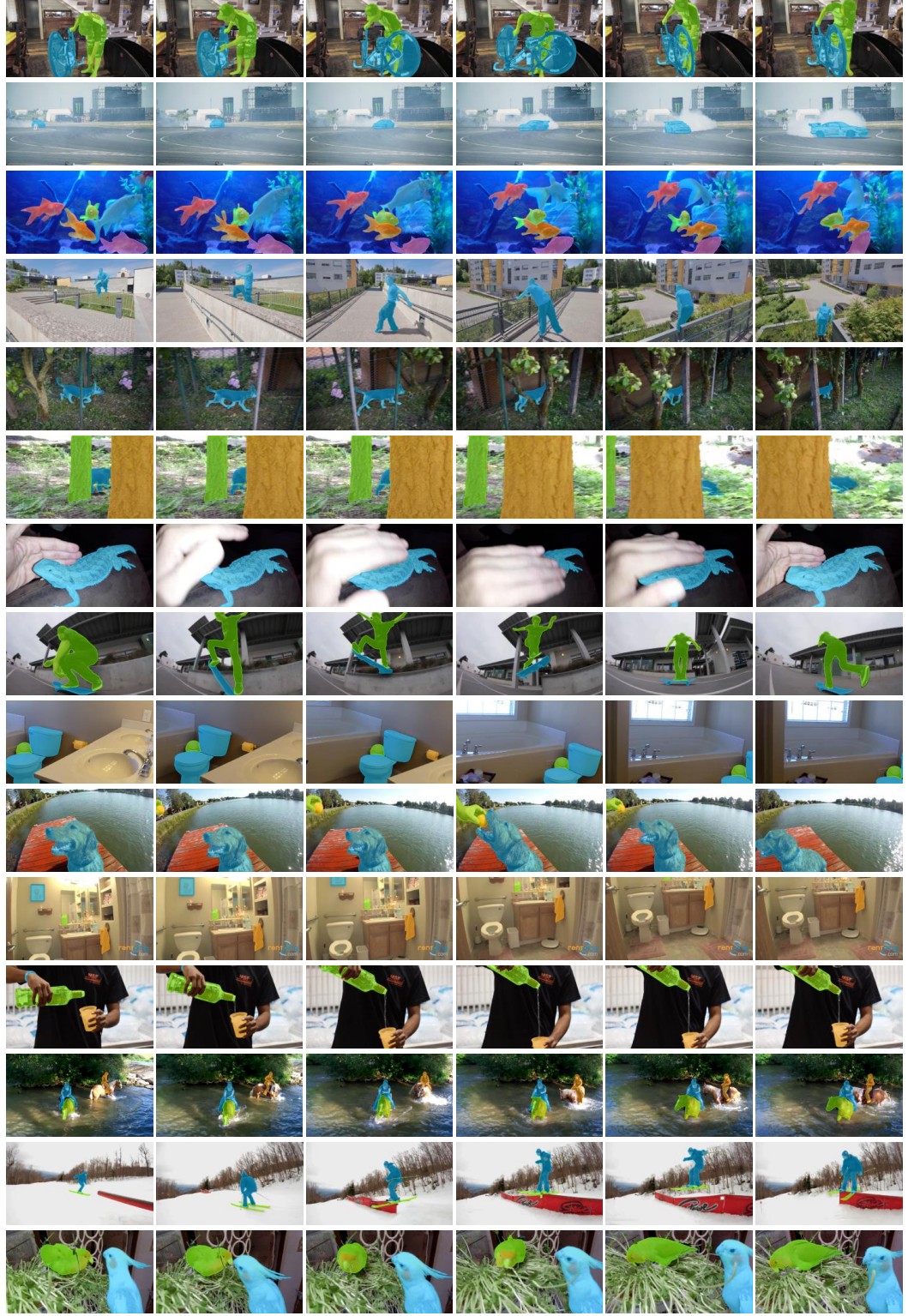

Figure 12: Visualization of video object segmentation.

