# OpenReview forum: "Matcher: Segment Anything with One Shot Using All-Purpose Feature Matching"
_ICLR.cc/2024/Conference — ICLR 2024 poster_

### Official Review · Reviewer_TD84 · 2023-10-28

**Soundness:** 4 excellent
**Presentation:** 4 excellent
**Contribution:** 4 excellent
**Rating:** 8
**Confidence:** 5

**Summary:**

This paper introduces Matcher, a framework that leverages pre-existing vision foundation models to tackle diverse perception tasks. The Matcher framework encompasses three key components: bidirectional matching for precise matrix correspondences extraction, a range of sample prompt designs encompassing part-level, instance-level, and global-level prompts, and controllable mask generation through instance-level matching. Through extensive experimentation on multiple benchmarks, such as COCO-$20^i$ and LVIS-$92^i$, the proposed method's efficacy is demonstrated. Furthermore, quantitative results highlight Matcher's ability to handle images in real-world scenarios, showcasing its open-world generality and flexibility.

**Strengths:**

1. The paper is well-written and presents its ideas in a clear manner, making it easy for readers to follow the proposed framework.

2. The idea of Matcher model is straightforward and practical. The three key components within the Matcher framework are not only effective but also highly efficient. (1) Bidirectional matching for precise matrix correspondences extraction; (2) A range of sample prompt designs encompassing part-level, instance-level, and global-level prompts; (3) Controllable mask generation through instance-level matching.

3. The paper demonstrates good performance not only on standard one-shot benchmarks but also on a Video Object Segmentation (VOS) benchmarks.

**Weaknesses:**

1. One crucial aspect of achieving high performance in the Matcher framework heavily relies on the utilization of DINO-V2 for accurate correspondence matrix extraction. However, it is worth noting that this approach may involve assembling multiple foundation models, potentially leading to a trade-off between accuracy and efficiency.

2. In light of recent research [1], it has been suggested that Stable Diffusion models offer a promising alternative for accurate correspondence matrix extraction, which has also been validated in the context of image-matching tasks. Consequently, it becomes pertinent to compare the performance of DINO-V2, the current method employed in the Matcher framework, with a diffusion-based correspondence extraction approach. Such a comparative analysis would enable a comprehensive evaluation of the two methods, shedding light on their respective capabilities within the Matcher framework and determining which approach yields superior results.

[1] Tang et al. Emergent Correspondence from Image Diffusion, in NeurIPS 2023.

**Questions:**

The running speed of the proposed method in terms of efficiency is an important aspect to consider. It would be valuable to compare the efficiency of the proposed method with related works to assess its performance in this regard. By conducting a comparative analysis, we can gain insights into how the proposed method fares in terms of running speed and efficiency when compared to existing approaches in the field.

---

> ### Author Response · Authors · 2023-11-15
> **Response to reviewer TD84**
>
> We thank you for your comments and the approval of our method and performance. In addition to the general response, we address your concerns here.
> ___
> > W1: Trade-off between accuracy and efficiency.
>
> Matcher is a flexible framework that can deploy different models. The tables below show that Matcher still achieves good performance when utilizing smaller DINOv2 and SAM. And we can use smaller Semantic-SAM (SwinT) [1] to replace SAM and achieve higher efficiency.
>
> |Methods|SAM|DINOv2|Param (M)|LVIS-92$^i$|FSS-1000|
> |-|-|-|-|-|-|
> |SegGPT|-|-|307|17.5|85.6|
> |Matcher|Base|Base|180|28.6|85.3|
> | | Large|Base|399|29.9|85.7|
> | | Large|Large|617|30.4|86.3|
> | | Huge | Large | 945 |31.4| 87.0|
>
>
> | | COCO-20$^i$ | LVIS-92$^i$ | FSS-1000 | PACO-Part |
> |-|-|-|-|-|
> |SAM | 52.7 | 31.4 | 87.0 | 32.7 |
> |Semantic-SAM|51.1|30.1|87.5|36.0|
>
> [1] Semantic-SAM: Segment and Recognize Anything at Any Granularity
>
> > W2: Comparison of the performance between DINOv2 and Stable Diffusion within the Matcher framework.
>
> We compared Stable Diffusion (SD) and DINOv2 on FSS, and the experimental results show that SD can achieve comparable performance to DINOv2, both models outperforming MAE and CLIP. Our experiments align with [1] and obtain similar conclusions. We hope *Matcher can serve as an evaluation criterion for VFMs, assessing the performance of different vision models from a general segmentation perspective*.
>
> | model | mIoU |
> | --- | --- |
> | MAE | 71.9 |
> | CLIP | 77.4 |
> | Stable Diffusion | 86.1 |
> | DINOv2 | 87.0 |
>
> [1] Tang et al. Emergent Correspondence from Image Diffusion, in NeurIPS 2023.
>
> > Q1: Running speed.
>
> Matcher aims at scenarios that **require high segmentation quality**, such as automatic annotation. In these scenarios, the speed of the Matcher is acceptable.
>
> In addition, the speed of Matcher can be improved through various methods such as model lightweight and parallel computing.

---

> > ### Author Response · Authors · 2023-11-21
> > **Happy to provide additional clarification**
> >
> > We sincerely thank you again for your great efforts in reviewing this paper. We have gone through your points one-by-one and tried to address them carefully. Please don’t hesitate to let us know if you have any further questions.

---

> > > ### Comment · Reviewer_TD84 · 2023-11-21
> > > **Keep my original score**
> > >
> > > I appreciate your counter-argument, it has effectively addressed my reservations and I shall maintain my evaluation.

---

### Official Review · Reviewer_3ZxQ · 2023-11-01

**Soundness:** 2 fair
**Presentation:** 3 good
**Contribution:** 3 good
**Rating:** 6
**Confidence:** 5

**Summary:**

This paper introduces Matcher,  which utilizes pre-trained vision foundation models for various perception tasks without requiring specific fine-tuning or training. Matcher is capable of segmenting images using in-context examples, demonstrating its adaptability and proficiency across multiple segmentation tasks.

**Strengths:**

1. The paper presents a good way to generate prompts for the visual fundation model Segment Anything Model (SAM). It is exquisite to plug-in and play with another pretrained vision model.
2. The method do not require any training or fine-tunning.
3. Extensive experiments and get reasonable results on many tasks.

**Weaknesses:**

1. Since there are a lot of engineering technique in the paper, one weakness is no open source code available. It would be nice to make the code public in the supplementary or in a public github repo. I would raise soundness score if the code is published.
2. Some of the comparison in experiments section is not fair. For example, all the results from SegGPT[1] are used a smaller ViT-L backbone. Even though SegGPT's training data includes the Coco dataset, it's important to note that these pre-trained models like DINOv2 or CLIP have been pre-trained on much larger datasets. So it is hard to say if it is fair to compare such generalist model, but at least we should compare with model of the same size.
3. Some of the details of implementation is missing. It lacks some of intresting abblation study. I will mention in quesions section below.

[1] X, Wang et al. SegGPT: Segmenting Everything In Context. ICCV, 2023

**Questions:**

1. Since SAM has already been trained on very large dataset, why not use SAM's encoder to generate propmt for itsself? If SAM has bad semantics, any intuision why is it?
2. How exactly do you use DINOv2? Where layers' output do you use as the feature?
3. How do you decide the number of clusters used do you use for Robust Prompt Sampler? I sould assume different objects may need different number of clusters. Any ablative study and explanation of these?
4. It would be nice to show some visualizations of different examples how exactly the points prompts are filtered by Patch-Level Matching, Robust Prompt Sampler and instance level matching.
5. To my understanding, the mistake made by Matcher depends on how good the semantics of the pretrained model to generate points and the mistake made by SAM itself. One interesting ablation study would be random sample same number of positive points on SAM as the number of points Macher generated, and take a look at the uper bound performance Matcher could achieve.
6. How does few-shot(>1) segmentation work?
7. For few shot segmentation, are the mistakes from localizations or bad mask shape? Some metric of localizaiton would be nice or visualizing more failure cases.

---

> ### Author Response · Authors · 2023-11-15
> **Part 1 of Response to reviewer 3ZxQ**
>
> We thank you for your comments and the approval of our method and performance. In addition to the general response, we address your concerns here.
> ___
> > W1: Since there are a lot of engineering technique in the paper, one weakness is no open source code available. It would be nice to make the code public in the supplementary or in a public github repo. I would raise soundness score if the code is published.
>
> We have submitted part of the core code in the supplementary materials to provide reviewers with a better understanding of the details. And we promise to open-source our code.
>
> > W2: Comparison between SegGPT and Matcher with the same size.
>
> We have updated the comparison between SegGPT[1] and Matcher with different model sizes in Table 8(a).
>
> The table below shows the results of Matcher when using VFMs with different model sizes. When using SAM base and DINOv2 base, Matcher still performs well on various datasets and achieves better generalization performance on LVIS-92i than SegGPT. Besides, as the model size increases, Matcher can continuously improve performance.
>
> We believe that utilizing VFMs effectively is key to exploring new paradigms in visual perception. While performance is important, we want to claim that Matcher is the first segmentation framework to effectively transfer VFMs to various segmentation tasks without any training.
>
> |Methods|SAM|DINOv2|Param (M)|LVIS-92$^i$|FSS-1000|
> |-|-|-|-|-|-|
> |SegGPT|-|-|307|17.5|85.6|
> |Matcher|Base|Base|180|28.6|85.3|
> | | Large|Base|399|29.9|85.7|
> | | Large|Large|617|30.4|86.3|
> | | Huge | Large | 945 |31.4| 87.0|
>
> > Q1: Since SAM has already been trained on very large dataset, why not use SAM's encoder to generate propmt for itsself? If SAM has bad semantics, any intuision why is it?
>
> SAM's encoder fails to match multiple objects/parts accurately, resulting in poor performance. Concurrent PerSAM [1] only utilizes SAM's representation for matching, which aligns with our findings based on experimental results (refer to Tables 1, 2, and 3 in the paper).
>
> The table below shows the linear probing experiments on ImageNet-1K using different pre-trained models (conducted by us or reported in other papers). SAM performs worse compared to DINOv2. We hypothesize that SAM's category-agnostic segmentation pre-training leads the network to learn the appearance and structural information of objects rather than acquire good semantic representations.
>
> |Methods|Model Size|Linear Probing|
> |-|-|-|
> |SAM|Huge|41.2|
> |DINOv2|Large|86.3|
>
> [1] Personalize Segment Anything Model with One Shot.
>
> > Q2: How exactly do you use DINOv2? Where layers' output do you use as the feature?
>
> We use the features from the last layer of DINOv2.
>
> > Q3: How do you decide the number of clusters used do you use for Robust Prompt Sampler? I sould assume different objects may need different number of clusters. Any ablative study and explanation of these?
>
> We have updated the ablation study of the number of clusters in Table 8(b).
>
> We set the number of clusters as eight and apply an equal number to all scenes. Eight clusters can perform well, so we did not conduct a detailed ablation study. We supplemented this experiment and the results are as follows:
>
> |Number of clusters|FSS-1000|
> |-|-|
> |4|78.85|
> |6|83.26|
> |8|87.0|
> |10|87.23|
> |12|86.9|
>
> > Q4: It would be nice to show some visualizations of different examples how exactly the points prompts are filtered by Patch-Level Matching, Robust Prompt Sampler and instance level matching.
>
> We have added the visualizations of different examples of how exactly the points prompts are filtered by PLM, RPS, and ILM in Figure 7.
>
> > Q5: The upper bound performance Matcher could achieve.
>
> Thank you for your helpful suggestion, which helps us gain further insights into Matcher. We have added the upper bound analysis of Matcher in Table 7(d) of the draft.
>
> We conducted experiments on four different datasets and found that the upper bound of Matcher consistently outperforms the current performance on all datasets by a large margin. This indicates that the Matcher framework has more potential. Therefore, Matcher can serve as an effective evaluation criterion for VFMs, assessing the performance of different vision models from a general segmentation perspective. Based on the advantage, Matcher can contribute to developing VFMs.
>
> | | COCO-20$^i$ | LVIS-92$^i$ | PACO-Part | FSS-1000|
> |-|-|-|-|-|
> |Upper Bound|	83.6|75.4|67.5|	93.1|
> |Matcher|52.7|31.4|32.7|87.0|
>
> > Q6: How does few-shot(>1) segmentation work?
>
> We have added the analysis of how few-shot(>1) segmentation works in Appendix D of the draft.
>
> In the few-shot setting, we concatenate multiple references' features and match them with the target image in the PLM. The remaining process is the same as the one-shot setting. Multiple samples provide richer visual details, enabling more accurate matching results and reducing outliers, resulting in performance improvement (the response for Q7 validates this).

---

> ### Author Response · Authors · 2023-11-15
> **Part 2 of Response to reviewer 3ZxQ**
>
> > Q7: For few shot segmentation, are the mistakes from localizations or bad mask shape? Some metric of localizaiton would be nice or visualizing more failure cases.
>
> Based on the upper bound analysis in Q5, we think performance is primarily limited by incorrect localization (i.e., outliers).
>
> We use the modified **Purity** and **Coverage** as the metrics of localization.
>
> **Purity**=number of the points matched with the target GT / number of the points within the target GT
>
> **Coverage**=number of the points matched with the target GT / number of all matched points
>
> A **smaller Coverage** indicates **more outliers** within the matched points, resulting in more false-positive masks.
>
> The tables below show the changes in **Purity** and **Coverage** as the shot increases when using forward matching and bidirectional matching (ours). When using only forward matching, a large number of outliers are produced as the reference increases, resulting in many false-positive masks. The proposed bidirectional matching strategy effectively mitigates this problem. However, as the reference increases, limited by the capacity of the image encoder, outliers will still inevitably increase, restricting further performance improvement.
>
> Forward matching:
> | | 1| 2 | 3 | 4 | 5 | 6 | 7 | 8 | 9 |
> | -| -| - | - | - | - | - | - | - | - |
> |Purity| 53.4 | 74.7 |83.7|90.6|93.3|94.9|95.8|97.1|97.8|
> |Coverage |47.7|37.0|31.0|26.1|22.2|18.9|18.6|18.4|17.0|
>
>
> Bidirectional matching (Ours):
> | | 1| 2 | 3 | 4 | 5 | 6 | 7 | 8 | 9 |
> | -| -| - | - | - | - | - | - | - | - |
> |Purity|42.0 | 55.2 | 59.6 | 64.5 | 64.8 | 66.1 | 66.8 | 66.7 | 67.0|
> |Coverage |65.6|62.9|61.1|60.7|58.6|57.9|59.0|58.6|57.6|
> |mIoU|52.7|55.7|57.6|60.3|60.1|62.4|58.8|59.1|59.0|
>
> The results indicate that the design of the Matcher is crucial because relying solely on DINOv2 leads to a large number of outliers and bad performance. Additionally, the proposed Purity and Coverage can serve as metrics for evaluating the general segmentation performance of different VFMs.

---

> > ### Author Response · Authors · 2023-11-21
> > **Happy to provide additional clarification**
> >
> > We sincerely thank you again for your great efforts in reviewing this paper. We have gone through your points one-by-one and tried to address them carefully. Please don’t hesitate to let us know if you have any further questions.

---

### Official Review · Reviewer_zVyp · 2023-11-01

**Soundness:** 3 good
**Presentation:** 3 good
**Contribution:** 2 fair
**Rating:** 5
**Confidence:** 4

**Summary:**

This paper proposes a general perception paradigm Matcher that utilizes off-the-shelf VFMs to address various segmentation tasks, such as few-shot/one-shot semantic/part segmentation. The VFMs in Matcher include DINOv2 as s feature encoder and SAM a promptable segmenter. Matcher involves no training, and achieves good results on several datasets.

**Strengths:**

* Performance is good: Matcher achieves the highest scores over different segmentation benchmarks.
* Every designed component is reasonable and works well from ablation studies.
* Matcher doesn't need training. This improves its efficiency.

**Weaknesses:**

* This work seems like a combination of existing vision foundation models and a set of engineering tricks. The pipeline can be summarized as "encode by DINOv2 -> select prompt (matching and sampling) -> prompt SAM -> select mask". Although the result proves its effectiveness, the paper is more like a technical report and lack academic insight.

* From the ablation in table 7(a), the performance of Matcher is largely influenced by different image encoders: MAE (18.8%), CLIP (32.2%), DINOv2 (52.7%). Does this mean the the selection of encoders would outweigh all the designs in Matcher?

* More ablation is needed for the selection of promptable segmenters, such as SEEM and Semantic-SAM. As the author claims Matcher as a general paradigm, does the designs still work on different segmenters?

* What would happen if the test image contains no reference concept? Can Matcher solve this situation, or output a false positive?

**Questions:**

See weakness.

---

> ### Author Response · Authors · 2023-11-15
> **Part 1 of Response to reviewer zVyp**
>
> We thank you for your comments and the approval that our method is efficient and works well. In addition to the general response, we address your concerns here.
> ___
> > W1: Although the result proves its effectiveness, the paper is more like a technical report and lack academic insight.
>
> Exploring new capabilities of foundation models to handle various downstream tasks is an emerging research trend. In the NLP field, in-context learning and chain-of-thought are extensively studied to improve the complex reasoning capability of LLMs. Similarly, exploring the application potential of vision foundation models (VFMs) is non-trivial, but related research in the CV field lags far behind NLP. Therefore, our **novelty** lies in exploring a new visual research paradigm: investigating the capabilities of off-the-shelf VFMs to accomplish various complex segmentation tasks.
>
> To contextualize the results in this paper, we now summarize our **insights**:
>
> - **Investigating the capabilities of VFMs.** Utilizing VFMs to address various tasks is becoming a research trend [1,2,3,4]. We transfer off-the-shelf VFMs into a general segmentation model, which is meaningful (**supported by Reviewer PEjB**).
> - **Proposing a proper outlier detection pipeline.** We transform the in-context segmentation task into a matching problem between the objects in the reference and target images and utilize off-the-shelf VFMs to achieve this goal. Simply concatenating these models will **produce many outliers** (see Figure 7) and **perform badly.** Outlier detection is a **challenging academic problem [5].** The outliers are often detected with algorithms, e.g. RANSAC given a predefined loss function. In Matcher, we consider **proposing a proper outlier detection pipeline is important and nontrivial.**
> - **Impressive performance**. Matcher has achieved impressive performance across multiple segmentation tasks, and extensive ablation studies have confirmed that the proposed components are effective and necessary.
> - **Serve as an evaluation criterion for VFMs.** Matcher has a high upper bound of performance. It can serve as an evaluation criterion for VFMs, assessing the performance of different vision models from a general segmentation perspective. Based on the above contributions, Matcher is helpful to the vision community and provides academic insight.
>
> Taken together, these contributions represent novel insights to the academic community.
>
> [1] A Tale of Two Features: Stable Diffusion Complements DINO for Zero-Shot Semantic Correspondence, NIPS23.
>
> [2] LMC: Large Model Collaboration with Cross-assessment for Training-Free Open-Set Object Recognition, NIPS23.
>
> [3] Prompt, Generate, then Cache: Cascade of Foundation Models makes Strong Few-shot Learners, CVPR23.
>
> [4] CNOS: A Strong Baseline for CAD-based Novel Object Segmentation.
>
> [5] Random sample consensus: a paradigm for model fitting with applications to image analysis and automated cartography. Communications of the ACM.
>
> > W2: Does the selection of encoders outweigh all the designs in Matcher?
>
> The table below represents the upper-bound analysis of Matcher (details can be found in the **response of Q5** for **3ZxQ**). The experiment shows that the Matcher framework has a high upper bound of performance. Different models exhibit performance differences. Table 7(a) validates that DINOv2 performs better. However, Table 4 shows that *simply using DINOv2 does not perform well*. *VFMs and the proposed Matcher components must work in synergy* and complement each other to be effective. Therefore, the designs in Matcher are very important.
>
> | | COCO-20$^i$ | LVIS-92$^i$ | PACO-Part | FSS-1000|
> |-|-|-|-|-|
> |Upper Bound|	83.6|75.4|67.5|	93.1|
> |Matcher|52.7|31.4|32.7|87.0|
>
> Like Semantic-SAM can replace SAM, we believe that with the development of the study of VFMs, there will be better models in the future that can achieve higher performance within Matcher. In addition, Matcher is a flexible training-free framework. It can be directly deployed with different VFMs without any training. Therefore, *Matcher can serve as an evaluation criterion for VFMs, assessing the performance of different VFMs from a general segmentation perspective without any training cost*. Therefore, we believe that Matcher is helpful to the vision community and provides academic insight.
>
> > W3: More ablation is needed for the selection of promptable segmenters.
>
> We have added the ablation study of different segmenters in Table 7(c) of the draft. The table below shows the results when using Semantic-SAM as Matcher’s segmenter. Semantic-SAM achieves comparable performance with SAM on four benchmarks. Because Semantic-SAM can output more fine-grained masks, it performs better than SAM on PACO-Part. *The results indicate that Matcher is a general segmentation framework*.
> | | COCO-20$^i$ | LVIS-92$^i$ | FSS-1000 | PACO-Part |
> |-|-|-|-|-|
> |SAM | 52.7 | 31.4 | 87.0 | 32.7 |
> |Semantic-SAM|51.1|30.1|87.5|36.0|

---

> > ### Author Response · Authors · 2023-11-15
> > **Part 2 of Response to reviewer zVyp**
> >
> > > W4: What would happen if the test image contains no reference concept? Can Matcher solve this situation, or output a false positive?
> >
> > Matcher checks if the input reference mask is empty and returns an empty output when the test image contains no reference concept.

---

> > > ### Author Response · Authors · 2023-11-21
> > > **Happy to provide additional clarification**
> > >
> > > We sincerely thank you again for your great efforts in reviewing this paper. We have gone through your points one-by-one and tried to address them carefully. Please don’t hesitate to let us know if you have any further questions.

---

### Official Review · Reviewer_PEjB · 2023-11-01

**Soundness:** 3 good
**Presentation:** 3 good
**Contribution:** 2 fair
**Rating:** 6
**Confidence:** 4

**Summary:**

A perception paradigm is introduced name Matcher. It can segment anything by using in-context learning and two pretrained large vision models, which are DINO and SAM. There are three effective components within Macther: Correspondence Matrix Extraction (CME), Prompts Generation (PG), and Controllable Masks Generation (CMG). The whole pipeline is without training and showcases state-of-the-art  performance among other generalist models.

**Strengths:**

1. The target task in this method is interesting to me. How be become a vision generalist is a hotspot in nowadays community. Matcher follows SegGPT to transfer off-the-shelf VFMs into a general segmentation model.

2. This work constructs several new benchmarks for one-shot or few-shot in-context segmentation. They are challenging and meaningful to future works.

3. The experiments demonstrate impressive generalization performance across various segmentation tasks.

**Weaknesses:**

1. The whole pipeline is too cumbersome to me. So many prompts seem unimportant from Table 7b in appendix. The final performance is significantly rely on the complicated ILM postprocessing from Table 4a.

2. The impressive performance is likely due to an engineering ensemble of models. For example, the post merging of masks seems like a sort of output ensemble. Improvement from ensemble is common in previous segmentation works.

3. I'm concerning about the inference time comparing to SegGPT, since Matcher needs to run the large-scale DINOv2 and SAM for every test image.

**Questions:**

Can Matcher segment a background concept just like SegGPT, such as blue sky or pavement? How does Matcher perform on more complex MOSE dataset for VOS?

---

> ### Author Response · Authors · 2023-11-15
> **Response to reviewer PEjB**
>
> We thank you for your comments and the approval of our target task. In addition to the general response, we address your concerns here.
> ___
> > W1: The whole pipeline is cumbersome.
>
> Utilizing VFMs to address various perception tasks is becoming a research trend. We transform the in-context segmentation task into a matching problem between the objects in the reference and target images and utilize off-the-shelf VFMs to achieve this goal. Simply concatenating these models will **produce many outliers** (see Figure 7) and **perform badly** (see Table 4). *Outlier detection is a challenging academic problem* [1]. The outliers are often detected with algorithms, e.g. RANSAC given a predefined loss function. In Matcher, we consider *proposing a proper outlier detection pipeline is important and nontrivial.*
>
> The proposed PLM, RPS, and ILM are targeted for solving this problems. Table 4 demonstrates that **these components need to work together** collaboratively to enable the VFMs to handle different segmentation tasks effectively. Considering the aforementioned challenges and difficulties, **the proposed Matcher framework is clear and reasonable.**
>
> [1] Fischler, Martin A., and Robert C. Bolles. "Random sample consensus: a paradigm for model fitting with applications to image analysis and automated cartography." *Communications of the ACM* 24.6 (1981): 381-395.
>
> > W2: The impressive performance is likely due to an engineering ensemble of models.
>
> As shown in Table 4, *simply concatenating these models does not perform well*. We analyze the reason and propose three reasonable and effective components to solve this challenge (see the response of W1).  *Impacted by the outliers, ensembling all masks directly will produce many false-positive masks, leading to poor performance without the proposed PLM and ILM*. The Matcher framework validates the possibility of using VFM to handle various segmentation tasks and provides an effective solution.
>
> > W3: Inference time compared to SegGPT.
>
> Matcher aims at scenarios that **require high segmentation quality**, such as automatic annotation. In these scenarios, the speed of the Matcher is acceptable.
>
> In addition, the speed of Matcher can be improved through various methods such as model lightweight and parallel computing.
>
> > Q1: Can Matcher segment a background concept just like SegGPT, such as blue sky or pavement? How does Matcher perform on more complex MOSE dataset for VOS?
>
> Matcher can segment a background concept like SegGPT [1]. We have added the visualization of Matcher for the quality of background concept segmentation in Figure 6.
>
> We want to make clear that **Matcher mainly focuses on one-shot semantic/part segmentation**. Additionally, Matcher can also handle VOS. The table below compares Matcher with concurrent PerSAM-F [2] on YouTube-VOS 2018 and MOSE. Matcher achieves better performance compared with PerSAM-F on both datasets.
>
> |Methods| | | YouTube | | | | | | | | MOSE | |
> |-|-|-|-|-|-|-|-|-|-|-|-|-|
> | | G | Js |	Fs | Ju | Fu | | | | | J&F | J | F |
> |SegGPT |74.7|75.1| 80.2|67.4|75.9| | | | |45.1|42.2|48.0|
> |PerSAM-F |54.4|53.9|56.4|50.7|56.6| | | | |23.3|19.8|26.8|
> |Matcher| 67.4| 69.1| 61.6 | 70.4| 68.3| | |	| | 33.4|28.6|38.3|
>
> [1] SegGPT: Segmenting Everything In Context.
>
> [2] Personalize Segment Anything Model with One Shot.

---

> > ### Author Response · Authors · 2023-11-21
> > **Happy to provide additional clarification**
> >
> > We sincerely thank you again for your great efforts in reviewing this paper. We have gone through your points one-by-one and tried to address them carefully. Please don’t hesitate to let us know if you have any further questions.

---

### Author Response · Authors · 2023-11-15
**General Response and Summary of Updates to Manuscript**

We thank the reviewers for noting that we address an important task (**PEjB**) with a straightforward training-free method (**zVyp**, **3ZxQ**, **TD84**) and achieve impressive generalization performance across various segmentation tasks (**PEjB**, **zVyp**, **3ZxQ**, **TD84**). First, we provide a high-level summary of the changes that we've made to the draft to address your feedback, and conclude with an overview of our key contributions.
___
Here is the summary of updates that we've made to the draft:

- Added the visualization of Matcher for the quality of background concept segmentation in Figure 6. (**PEjB**)
- Added the ablation study of different segmenters in Table 7(c). (**zVyp**)
- Added the comparison between SegGPT and Matcher with different model sizes in Table 8(a). (**3ZxQ**)
- Added the ablation study of the number of clusters in Table 8(b). (**3ZxQ**)
- Added the visualizations of different examples of how exactly the points prompts are filtered by PLM, RPS, and ILM in Figure 7. (**3ZxQ**)
- Added the upper bound analysis of Matcher in Table 7(d). (**3ZxQ**)
- Added the analysis of how few-shot(>1) segmentation works in Appendix D.  (**3ZxQ**)

To end this update, we discuss the contributions and insights concerned by ****PEjB**** and ****zVyp****.



**Novelty and Academic Insights**

Exploring new capabilities of foundation models to handle various downstream tasks is an emerging research trend. In the NLP field, in-context learning and chain-of-thought are extensively studied to improve the complex reasoning capability of LLMs. Similarly, exploring the application potential of vision foundation models (VFMs) is non-trivial, but related research in the CV field lags far behind NLP. Therefore, our **novelty** lies in exploring a new visual research paradigm: investigating the capabilities of off-the-shelf VFMs to accomplish various complex segmentation tasks.

To contextualize the results in this paper, we now summarize our **insights**:

- **Investigating the capabilities of VFMs.** Utilizing VFMs to address various tasks is becoming a research trend. We transfer off-the-shelf VFMs into a general segmentation model, which is meaningful (supported by Reviewer **PEjB**).
- **Proposing a proper outlier detection pipeline.** We transform the in-context segmentation task into a matching problem between the objects in the reference and target images and utilize off-the-shelf VFMs to achieve this goal. Simply concatenating these models will **produce many outliers** (see Figure 7) and **perform badly.** Outlier detection is a **challenging academic problem** [1]. The outliers are often detected with algorithms, e.g. RANSAC given a predefined loss function. In Matcher, we consider **proposing a proper outlier detection pipeline is important and nontrivial.**
- **Impressive performance**. Matcher has achieved impressive performance across multiple segmentation tasks, and extensive ablation studies have confirmed that the proposed components are effective and necessary.
- **Serve as an evaluation criterion for VFMs.** Matcher has a high upper bound of performance (see Table 7(d)). It can serve as an evaluation criterion for VFMs, assessing the performance of different vision models from a general segmentation perspective. Based on the above contributions, Matcher is helpful to the vision community and provides academic insight.

Taken together, these contributions represent novel insights to the academic community.

[1] Fischler, Martin A., and Robert C. Bolles. "Random sample consensus: a paradigm for model fitting with applications to image analysis and automated cartography." Communications of the ACM 24.6 (1981): 381-395.

---

> ### Author Response · Authors · 2023-11-22
> **Happy to provide additional clarification**
>
> Dear reviewers,
>
> We sincerely thank you again for your great efforts in reviewing this paper. As the Author-Reviewer discussion period is going to end soon, we want to know whether the reviewers have any additional questions based on our previous response. Please don’t hesitate to let us know if you have any remaining concerns.
>
> Thank you,
>
> The authors

---

### Meta-Review · Area_Chair_6Q67 · 2023-12-19

**Metareview:**

All the reviewers recognized the strength of the paper: it’s an innovative approach to take on transferring off-the-shelf vision foundation models (VFMs) into a general segmentation model; the work created a new challenging & meaningful benchmark for one-shot and few-shot in-context segmentation; the experiments show remarkable generalization performance across various segmentation tasks; the model demonstrated strong performance not only on standard one-shot benchmarks but also on video object segmentation (VOS) benchmarks.

Meanwhile, the reviewers also identified the weaknesses: the overall pipeline is deemed cumbersome and the final performance relies heavily on complicated instance-level matching (ILM) postprocessing; the impressive performance is attributed to an engineering ensemble of models rather than novel academic insight; the choice of image encoder might outweigh other designs in matcher as there is significant influence of different image encoders on performance; some comparisons in the experiments are not considered fair, particularly regarding the size of models and training datasets.

During the rebuttal, the authors provided good additional results and some of the reviewers raised their scores. The only reviewer R# zVyp, who remained slightly negative, unfortunately didn’t respond to the rebuttal even after the AC emailed the reviewer twice.

The AC feels that R# zVyp’s major concern, that the paper consists of a combination of foundation models and therefore lacks insight, is not valid as our community should be inclusive to new ideas and certainly foundation models were developed to be cornerstones for building more advanced/specialist models. The AC recommends “accept” as the paper demonstrated superior performance and the contribution is acknowledged by all three other reviewers.

**Justification For Why Not Higher Score:**

The paper consists of a good combination of off-the-shelf vision foundation models and got good results, but orals/spotlights are reserved for papers with greater impact. I don't feel that this paper reached that impact compared to others in my batch.

**Justification For Why Not Lower Score:**

The paper explored a new area with a lot of novel ideas and good results. Three of four reviewers recommended (marginal) accept. I think it will start a trend of using VFMs to do a lot of cool things.

---

### Decision · Program_Chairs · 2024-01-16

Accept (poster)